# Filopodia rotate and coil by actively generating twist in their actin shaft

Natascha Leijnse[1,5], Younes Farhangi Barooji[1,5], Mohammad Reza Arastoo [1], Stine Lauritzen Sønder[2], Bram Verhagen[1], Lena Wullkopf[3], Janine Terra Erler [3], Szabolcs Semsey[1], Jesper Nylandsted [2,4], Lene Broeng Oddershede [1], Amin Doostmohammadi [1✉] & Poul Martin Bendix [1✉]

Filopodia are actin-rich structures, present on the surface of eukaryotic cells. These structures play a pivotal role by allowing cells to explore their environment, generate mechanical forces or perform chemical signaling. Their complex dynamics includes buckling, pulling, length and shape changes. We show that filopodia additionally explore their 3D extracellular space by combining growth and shrinking with axial twisting and buckling. Importantly, the actin core inside filopodia performs a twisting or spinning motion which is observed for a range of cell types spanning from earliest development to highly differentiated tissue cells. Non-equilibrium physical modeling of actin and myosin confirm that twist is an emergent phenomenon of active filaments confined in a narrow channel which is supported by measured traction forces and helical buckles that can be ascribed to accumulation of sufficient twist. These results lead us to conclude that activity induced twisting of the actin shaft is a general mechanism underlying fundamental functions of filopodia.

[1] Niels Bohr Institute, University of Copenhagen, 2100 Copenhagen, Denmark. [2] Membrane Integrity, Danish Cancer Society Research Center, Strandboulevarden 49, 2100 Copenhagen, Denmark. [3] Biotech Research and Innovation Centre (BRIC), University of Copenhagen, Ole Maaløes Vej 5, 2200 Copenhagen, Denmark. [4] Department of Cellular and Molecular Medicine, Faculty of Health Sciences, University of Copenhagen, Blegdamsvej 3C, 2200 Copenhagen, Denmark. [5] These authors contributed equally: Natascha Leijnse, Younes Farhangi Barooji. ✉email: doostmohammadi@nbi.ku.dk; bendix@nbi.ku.dk

Mechanical interactions between cells and their environment are essential for cellular functions like motility, communication, and sensing. The initial contact formed by cells is mediated by F-actin rich filopodia which are highly dynamic tubular structures on the cell surface that allow cells to reach out and interact with their extracellular environment and adjacent cells[1].

Filopodia are present in a wide variety of cell types ranging from embryonic stem cells[2] to neuronal cells[3–6], they are important for cell migration during wound healing[7] and in cellular disorders such as cancer[8]. Recently, filopodia have been discovered to play critical roles in development by facilitating communication between mesenchymal stem cells[2] and during compaction of the early embryo[9].

Structurally, filopodia are formed as thin membrane protrusions (diameter between 100 and 300 nm) containing 10–30 bundles of actin filaments, which are cross-linked by molecules such as fascin[10]. Transmembrane integrins link the actin to the cell membrane while peripheral proteins like IBAR link the actin to the inner leaflet of the plasma membrane[1].

Filopodia can differ significantly in length from a few micrometers to tens of micrometers. Long specialized filopodia can be sub-categorized into: Cytonemes[11] which are involved in long-range cell signaling; tunneling nanotubes[2] involved in intercellular material exchange including cell-cell virus transmission[12]; and recently discovered airinemes[13] found on skin resident macrophages involved in pigment pattern formation during zebrafish development.

Despite the high diversity of mechanical functions carried out by filopodia there seem to be common characteristics which are preserved in all types of filopodia. These include typical traction forces of tens of piconewtons[14–18], growth and shrinkage[3], and bending or lever arm activity[19].

Growth and shrinkage of filopodia are regulated by actin polymerization at the tip[20] and myosin activity which contributes to retrograde flow[18,21]. In addition to this, a sweeping motion of the filopodial tip around the cellular base has been reported[3,22] and even rotational motion has been indicated in HEK293 cells[17] and neuronal cells[6]. However, these studies have not been able to decipher whether the filamentous actin inside the filopodium, in the following denoted as 'actin shaft', performs a spinning motion around its own axis or whether the filopodium performs a circular sweeping motion around the anchor point. Complex movements and helical buckling shapes have been reported together with simultaneous force generation[6,17,22–24], thus indicating build-up of torsional twist in the actin shaft of HEK293 cell filopodia. The dynamics of filopodial movement has been shown to be unaffected by inhibition of myosin II[24]. Knock-down of myosin Va and Vb, which are motors walking in a spiral path on actin filaments, was found to reduce the lateral movements of filopodia tips[6,25,26]. However, the generality and underlying mechanisms for all these modes of movement have remained enigmatic.

Here we show that twist generation in the actin shaft can explain several of the observed phenomena such as helical buckling, traction, and rotational movements of filopodia. We developed an advanced integrated optical tweezers and confocal microscopy assay to visualize the rotation of the F-actin shaft inside a filopodium. An optical trap is used to fix the filopodial tip and also used to attach a bead to the side of a filopodium. The bead is linked to the actin inside the filopodium via vitronectin-integrin bonds and hence reports about any axial rotation and retrograde flow of actin. The observed axial rotations of the bead are compared to results from tracking of filopodial tips of cells grown on glass or embedded in a collagen I matrix. Furthermore, helical buckles resulting from twist accumulation are detected in a number of cell types, on glass as well as embedded in collagen I. To test whether these buckles originate from membrane-induced compressional load[27] or from the accumulation of twist within the actin shaft resulting from the spinning, we quantify the forces acting along the actin shaft. The generality of these phenomena was assessed by measuring the traction forces of both naive pluripotent stem cells and terminally differentiated cells. Interestingly, despite the presence of traction forces delivered by filopodia in all cell types, we frequently observed helical buckling in the actin shaft which can be explained by build-up of torsional twist.

We identify a mechanism behind the twisting and consequent spinning of the actin core by modeling actin/myosin complexes inside the filopodial membrane as an active gel, showing that mirror-symmetry breaking and the emergence of twist are generic phenomena in a confined assembly of active filaments, suggesting the physical origin of the observed twisting motion in a variety of cell types.

## Results

**Filopodia and extracted membrane tethers can rotate, pull, and buckle**. Filopodia show rich and complicated dynamics that is connected to remodeling of their actin shaft. On the order of minutes, they undergo significant movement and reshaping. Filopodial tips are often found to rotate[6,17,22,28] or perform a sweeping motion as shown for a EGFP Lifeact-7 expressing HEK293T cell on glass in Fig. 1a and b and Supplementary Movie 1 and Supplementary Fig. 1. Bending or coiling are frequently observed phenomena, shown in Fig. 1c and d for a KP$^{R172H}$C cell (cyan, mEmerald Lifeact-7, in the following, denoted as KPC) embedded in a matrix containing 4 mg/ml collagen I (Supplementary Fig. 2 and Supplementary Movie 2). The traction force generated by filopodia is examined by extracting a new filopodium from the plasma membrane using an optically trapped bead (Fig. 1e–g), and the trapping of the tip also reveals the frequent formation of buckles observed along the filopodium. Buckle formation additionally took place without prior imaging with the confocal scanning laser, thus excluding that buckling is an artifact of the cell reacting to the laser illumination, see Supplementary Fig. 3.

The rotating motion of a filopodium and the associated shape changes and generation of traction, shown in Fig. 1, indicate that the actin shaft contains twist that can be converted into buckling and traction. We therefore next sought to investigate the possible generation of torque by an actomyosin system confined within a tether extracted from a living cell.

**The actin shaft within extracted membrane tethers performs a spinning motion**. Filopodia-like structures were formed by the extraction of membrane tethers from the cell surface by using optical tweezers[17–19,29–31]. Depending on the time of observation F-actin can be found to be present inside pulled membrane tethers which subsequently exhibit similar behavior as native filopodia[3,4,8,17,32,33]. To visualize and quantify any possible actin shaft rotation we use the setup schematically shown in Fig. 2a. A tether, seen as an artificially extended filopodium, was extracted from an EGFP Lifeact-7 expressing cell (as a marker for F-actin[34]) using an optically trapped vitronectin (VN) coated bead ($d = 4.95\,\mu m$). Vitronectin binds to transmembrane integrins which can serve as a link between the bead and the cytoskeletal filamentous actin. Following extraction of the membrane tether the bead was immobilized on the glass surface of the chamber to hold the hereby newly formed filopodium in place. We then used the optical trap to attach another smaller VN coated red labeled bead ($d = 0.99\,\mu m$, in the following denoted as 'tracer bead') to

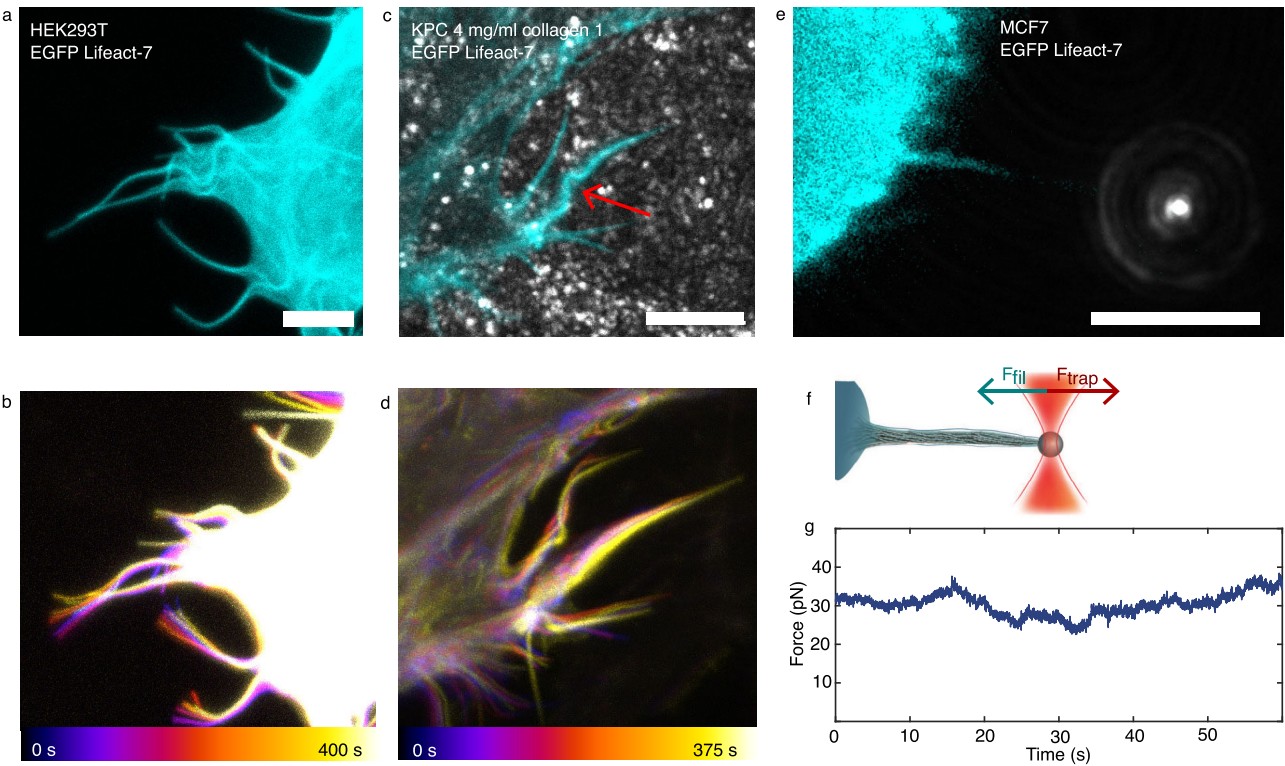

**Fig. 1 Free and confined filopodia show rich dynamics that can lead to bending, buckling, coiling, shortening, and pulling. a, b** The tips of free filopodia rotate. Confocal Z-stacks of a HEK293T cell (cyan, EGFP Lifeact-7) grown on glass. **a** Confocal Z-projection of a single Z-stack. Scale bar is 5 $\mu$m. **b** Overlay of Z-projections at 11 consecutive time points (total time is 400 s, color coded from blue to white). **c, d** Filopodia confined by collagen frequently exhibit buckles. Confocal Z-stacks of a KPC cell (cyan, EGFP Lifeact-7) grown in 4 mg/ml collagen I (gray, reflection). The red arrow marks filopodial bending regions. **c** Confocal Z-projection of a single Z-stack. Scale bar is 5 $\mu$m. **d** Overlay of Z-projections at 6 consecutive time points (total time is 375 s, color coded from blue to white). **e–g** Membrane tethers extracted from cells fill up with F-actin (actin shaft), are highly dynamic, and behave like filopodia. **e** Confocal image of a tether extracted from a MCF7 cell (cyan, EGFP Lifeact-7) using an optically trapped vitronectin coated $d = 4.95$ $\mu$m bead (gray, reflection). Scale bar is 5 $\mu$m. **f** Schematics of the setup for tether extraction: the force exerted by the optical trap (orange laser beam profile) holds a bead (gray) in place. This bead is used to extract a membrane tether from the cell (cyan). The tether exerts a pulling force on the bead (green arrow) towards the cell body, counteracted by the trap force (red arrow). The force exerted by the tether is measured by tracking the bead's displacement relative to the initial center of the trap. Over time, the tether begins to bend, coil, and hence shortens and exerts a traction force on the trapped bead. **g** Holding force as a function of time exerted by the tether from (**e**) versus time. Source data are provided as a Source Data file. Brightness/contrast of the color channels (reflection and fluorescence) were adjusted individually.

the tether and acquired Z-stacks over time using a confocal microscope. The VN coating on the tracer bead ensures that the bead binds to transmembrane integrin proteins which can interact with actin on the cytosolic side of the membrane. As the actin filaments are known to undergo retrograde flow, so will the integrins and the bead attached to the integrins will move along. Hence, this assay should report the movement of the actin shaft inside the tether.

Results from this assay are shown in Fig. 2b and Supplementary Fig. 4 for a bead attached to a tether from a HEK293T cell ('cell 1'). Figure 2c–g and Supplementary Movie 3 and 4 show bead rotation data for 'cell 2', an uninduced MCF7-p95ErbB2 breast carcinoma cell[35,36].

As shown in Fig. 2b–d, f Supplementary Fig. 4, and Supplementary Movie 3 and 4, the tracer bead moves towards the cell body of the HEK293T and the uninduced MCF7-p95ErbB2 cell as expected due to the retrograde flow. The transport of the tracer bead along the filopodium was at the speed of $150 \pm 131$ nm/s, see Supplementary Table 1, and is consistent with retrograde flow of actin within the filopodium. This behavior was also observed for MCF7 cells not expressing the p95ErbB2 receptor, see further examples for MCF7 and HEK293T cells in Supplementary Fig. 5.

We used a custom-written MATLAB program to subtract the constant sideway movement of the filopodium and thereby align

the axis of the filopodium in the YZ-plane (see Supplementary Fig. 6 and 7). This revealed that the bead furthermore undergoes a spiraling motion around the filopodium. We isolated the rotation of the bead around the axis of the actin shaft (Fig. 2d, e) and found it to be clounterclockwise as seen from the tip towards the cell body. The rotation frequency of the VN coated tracer bead attached to 'cell 2' in Fig. 2c was 0.002 Hz as obtained from a power spectral analysis (Fig. 2g). Supplementary Table 1 shows additional tracer bead rotation frequencies measured for beads on filopodia from HEK293T, MCF7, and uninduced MCF7-p95ErbB2 cells.

The measured spinning of the actin core is also expected to result in the rotational motion of the filopodial tip. Therefore, we next tracked the three-dimensional movement of free filopodia and quantified their angular motion.

**The tips of filopodia rotate with a similar angular velocity as the spinning of the actin shaft.** We tracked the motion of F-actin labeled filopodia (via EGFP Lifeact-7 expression) in cells grown on glass slides, Fig. 3a, and cells embedded in collagen I gels, Fig. 3e. The tracking of filopodia was performed using a custom-written MATLAB algorithm which allows tracking free tips of filopodia during their growth and shrinkage. The algorithm

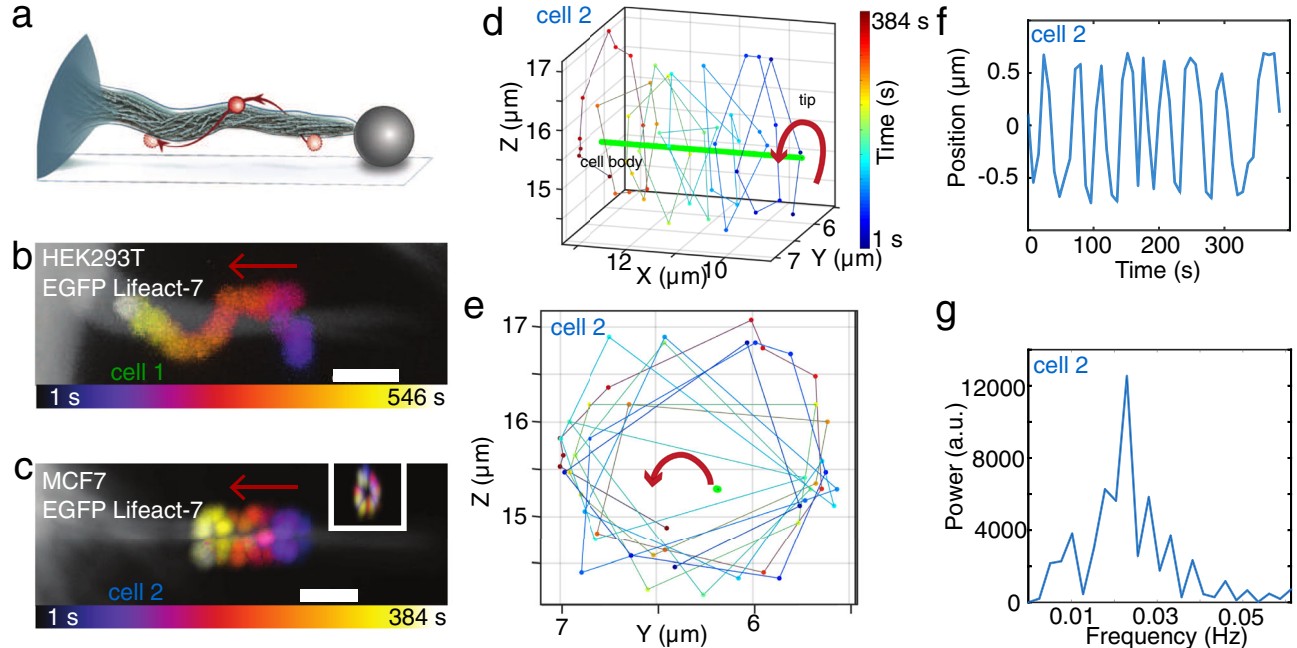

**Fig. 2 The actin shaft within filopodia performs a spinning motion. a** Assay to visualize internal F-actin rotation in filopodia: A tether of a cell on a glass slide (gray) is extracted from an EGFP Lifeact-7 expressing cell (cyan) using an optically trapped vitronectin coated bead (gray, $d = 4.95\ \mu m$). After tether extraction, the bead is attached to the glass surface of the sample such that the tether is held in place even when the trap is turned off. A tracer bead coated with vitronectin (red, $d = 0.99\ \mu m$) is attached to the tether using the optical trap. The tracer bead binds indirectly to the filamentous actin inside the tether via transmembrane integrins. After the tracer bead is attached, the trap is turned off and confocal $Z$-stacks are acquired at consecutive time points revealing the spinning of the F-actin shaft inside the filopodium. **b, c** Confocal $Z$-projections of a tether from a HEK293T cell (**b**, cell 1, gray, EGFP Lifeact-7) and a not activated MCF7-p95ErbB2 cell (**c**, cell 2, gray, EGFP Lifeact-7) at consecutive time points where an attached tracer bead rotates counterclockwise around the actin shaft from tip towards the cell body (color coded from yellow to red). Scale bars are 2 $\mu m$. Red arrows show direction of motion from tip towards cell body. The inset in (**c**) shows images of the tracer bead position at all time points in $YZ$ view. **d, e** 3D trajectory of the tracer bead (from blue to yellow, same time scale as in (**c**)) moving counterclockwise around the filopodium (green) towards the cell body in $XYZ$ (**d**) and $YZ$ (**e**) view. **f** Tracer bead position as a function of time for cell 2. **g** The rotation frequency of the tracer bead and thus the actin shaft rotation frequency is found to be 0.02 Hz, as obtained from the peak of the power spectrum. Source data are provided as a Source Data file.

allowed us to compensate for filopodial drift caused by the migration of the cell and by possible lateral sliding of the filopodium.

We acquired confocal $Z$-stacks of cells placed on glass or embedded within 4 mg/ml collagen I gels for approximately 5 min. Figure 3a, e shows an overlay of $Z$-projections of a filopodium from a MCF7 cell on glass and a KP$^{fl}$C cell in 4 mg/ml collagen I, respectively, at different time points. The lateral movement in Fig. 3a, e has been subtracted such that filopodia at all times initiated at the same point, and thus their rotary motion could be tracked in three dimensions, see Gabor-filtered image of the filopodium in Fig. 3b shown in $XZ$ (left) and $YZ$ view (right) (Supplementary Fig. 7).

Since the length of a single filopodium varies over time, we chose a common $YZ$ plane close to the tip through which the filopodium crossed at all time points. Figure 3c, d and f, g shows that the filopodial tips rotate counterclockwise over time in the $YZ$ plane as seen from the tip towards the cell body for the MCF7 cell on glass and the KP$^{fl}$C cell in a collagen I matrix, respectively. We obtained the frequency (Fig. 3h) or angular velocities (Fig. 3i) for different cell types cultured on glass or in collagen. The tips of filopodia on the MCF7 and KP$^{fl}$C cells rotated with a frequency of 0.008 Hz and 0.004 Hz, respectively. Figure 3i compares mean angular velocities for HEK293T, MCF7, and uninduced MCF7-p95ErbB2 cells on glass and KP$^{fl}$C and KPC cells in collagen I matrices. Angular velocities of filopodial tips in 3D collagen I matrices are clearly lower than of free cells (Fig. 3i, inset) indicating that the extracellular fibers confine and slow

down the tip movement. The mean angular velocity values and standard deviations as well as the p-values can be found in Supplementary Table 2.

Of the 68 filopodial tips of cells placed on glass or embedded in collagen (see Fig. 3i and Supplementary Table 2) we found 51% to undergo a counterclockwise rotation (as seen from the tip towards the cell body) and 9% were clockwise. In 40% of the cases we could not clearly determine a handedness which is due to a complex behavior of filopodia which includes actin shaft rotation, retrograde flow, buckling and overall cell movement (See example trajectories in Supplementary Fig. 8).

We also detected filopodia rotation in naive pluripotent mouse embryonic stem cells (HV.5.1 mESCs) grown in 2i medium, see Supplementary Movie 5. These cells represent the earliest development and hence could indicate whether filopodia rotation is a general property of cells. These observations together with the data from terminally differentiated cells like Hepa 1–6 mouse hepatocytes (Supplementary Movies 6, 7), and invasive cancer cells such as MCF7-p95ErbB2, see Supplementary Table 2, show that filopodia rotation takes place in both, the earliest and late stages of development.

**Myosin activity affects filopodia rotation.** To shed light on the active mechanism leading to filopodia rotations we performed mRNA silencing of genes coding for molecular activity within the filopodium. In particular myosin V and myosin X motors have been reported to be associated with filopodia formation and activity

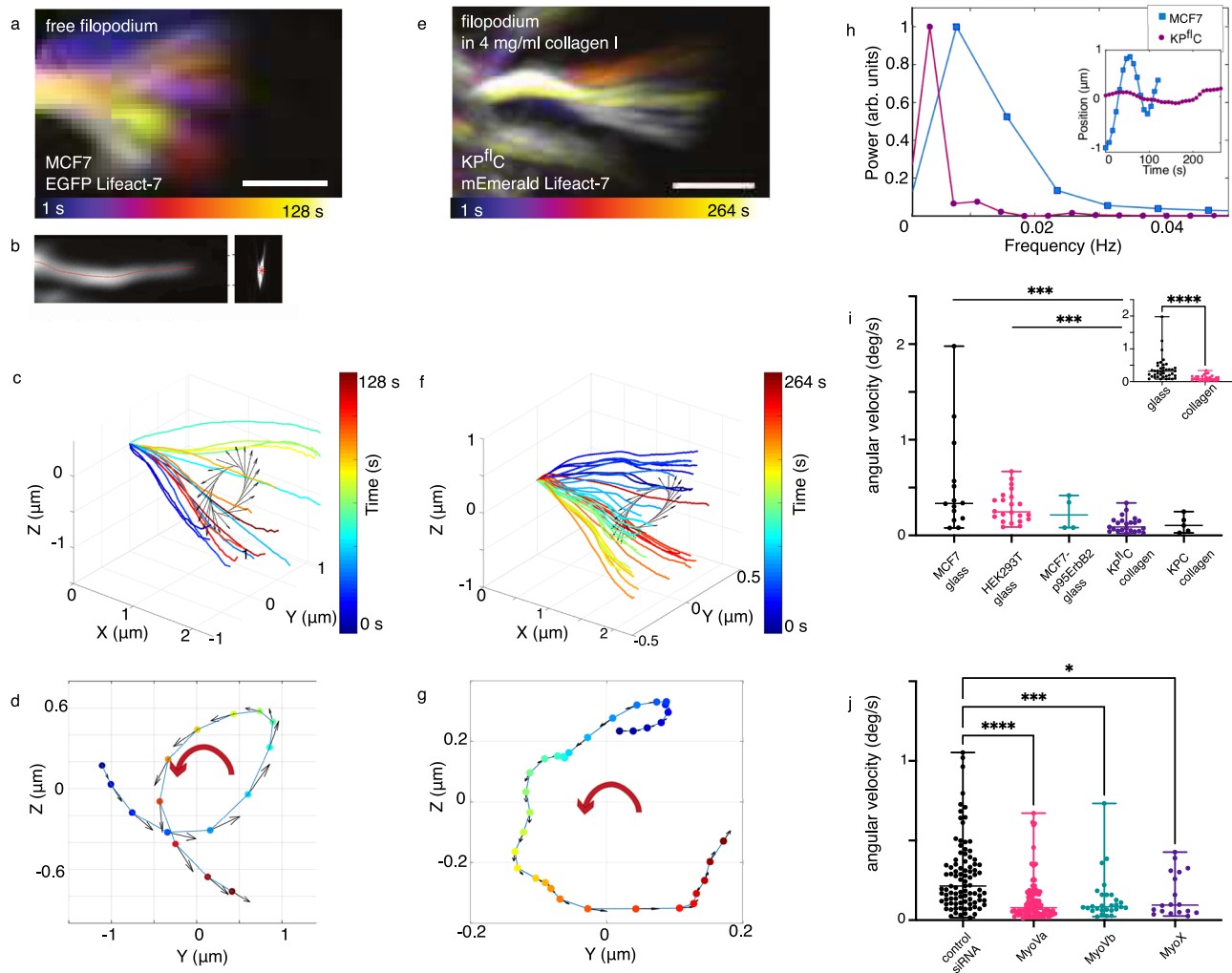

**Fig. 3 Tip rotations of filopodia from cells on glass or embedded within collagen I. a** Gabor-filtered *Z*-projection of filopodial tip movement of a EGFP Lifeact-7 labeled MCF7 cell imaged for 128 s. Scale bar, 1 μm. **b** *XZ* (left) and *YZ* (right), *Z*-projection showing the Gabor-filtered data (gray) overlayed with the traced filopodium (red). **c, d** Tip tracking of the filopodium from (**a**) in *XYZ* (**c**) and *YZ* view (**d**). **e** Gabor-filtered *Z*-projection of filopodial tip movement of an EGFP Lifeact-7 labeled KP^flC cell grown in 4 mg/ml collagen I for a total time of 264 s. Scale bar, 1 μm. **f, g** Tip tracking of the filopodium shown in (**e**) in *XYZ* (**f**) and *YZ* view (**g**). **h** The rotation frequency for the cell in (**a**) is 0.008 Hz and for the KP^flC cell shown in (**e**) is 0.004 Hz. Inset: Filopodial tip position versus time. **i** Angular velocities of filopodial tips from cells on glass (mean ± SD (deg/s), *N* filopodia: MCF7 (0.53 ± 0.53, *N* = 14), HEK293T (0.30 ± 0.16, *N* = 22), MCF7-p95ErbB2 (0.23 ± 0.18, *N* = 4)) and in collagen I (KP^flC (0.11 ± 0.08, *N* = 23), KPC (0.12 ± 0.09, *N* = 5)). Kruskal–Wallis with adjusted *p*-value by Dunn's test was used with significance set at *p* < 0.05. Adjusted *p*-values are 0.0003 and 0.0005, for MCF7/MCF7-p95ErbB2 and HEK293T/KP^flC, respectively. Scatter plot shows the median and the whiskers extend from minimum to the maximum values. Inset: pooled angular velocities of filopodia from cells grown on glass ((mean ± SD, *N* filopodia) 0.3722 ± 0.3539, *N* = 40) and in collagen I (0.1141 ± 0.08095, *N* = 28). Two-tailed Mann–Whitney test was used. Scatter plot shows the median and the whiskers extend from minimum to the maximum values, *p* < 0.0001. (**j**) Angular velocities after silencing myosins Va (mean ± SD (deg/s), *N* filopodia: (0.13 ± 0.14, *N* = 88), Vb (0.13 ± 0.15, *N* = 28), and myosin X (0.16 ± 0.14, *N* = 18), respectively, were compared to cells exposed to control siRNA (0.28 ± 0.22, *N* = 92). Kruskal–Wallis with adjusted *p*-value by Dunn's test was used with significance set at *p* < 0.05. Adjusted *p*-values are <0.0001, 0.0003, and 0.0171, for control siRNA vs MyoVa, Vb, and X, respectively. Scatter plot shows the median and the whiskers extend from minimum to the maximum values. Source data are provided as a Source Data file.

and have been shown to transport membrane proteins and vesicular content along the filopodium[6,37–41].

MCF7 cells were depleted for myosin Va, Vb, or myosin X by siRNAs which led to a significant reduction of the expression levels of these motors in MCF7 cells, see Supplementary Fig. 9. Expression of Lifeact GFP in MCF7 cells enabled us to track the filopodia in cells, depleted for myosins using siRNAs, and extract the mean angular velocity. MCF7 cells expressing Lifeact GFP, which were depleted for myosin Va by siRNAs, showed a significant reduction in the angular velocity of filopodia when compared to cells transfected with control siRNA (Fig. 3j). Furthermore, we also measured a reduction in the angular

velocity of filopodia in MCF7 cells depleted for myosin Vb or myosin X by siRNAs as compared to cells transfected with control siRNA (Fig. 3j). The chirality of the rotations was also affected by the mRNA silencing of myosin activity (see Supplementary Table 3) leading to higher degree of randomness for the orientation in cells depleted for myosin motors as compared to control cells.

Another molecular component important for filopodia function is the formin mDia1. mDia1 has been reported to respond to twist in actin filaments[42–44] and hence could be involved in twisting the actin shaft in filopodia. Expression of mEmerald-mDia1 in MCF7 cells showed that mDia1, besides being

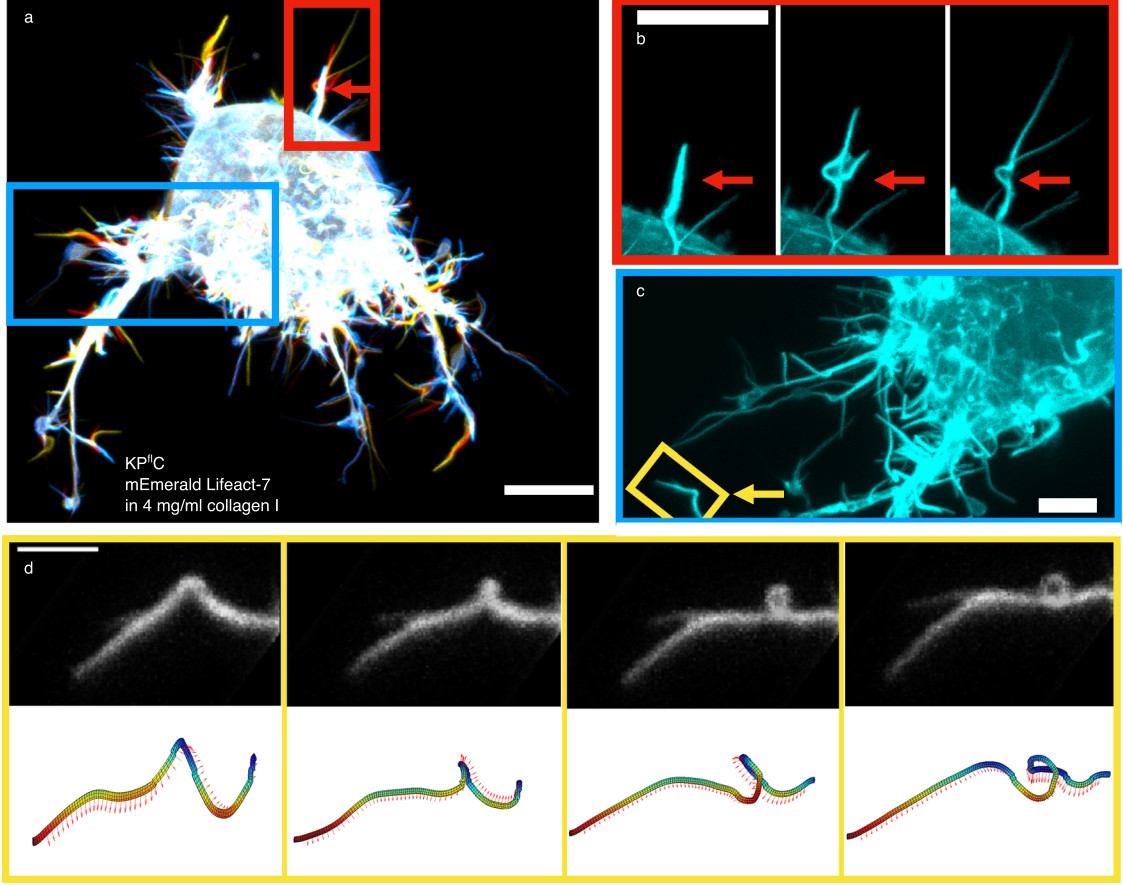

**Fig. 4 Filopodia of cells embedded in a collagen I matrix undergo helical buckling as a result of rotation and twist accumulation upon contact with the matrix. a** Overlay of 3 confocal *Z*-projections of a KP[fl]C cell (mEmerald Lifeact-7) embedded in 4 mg/ml collagen I at consecutive time points (color coded over 72 s). The red arrow highlights a region where a buckle forms. Scale bar is 10 $\mu$m. **b** Zoom-in on the red rectangular region from (**a**): individual images of the 3 time points show buckle formation. Scale bar is 10 $\mu$m, frame interval is 24 s. **c** Zoom-in on the blue rectangular region from (**a**) (rotated). The yellow rectangle highlights a bending filopodium. Scale bar is 5 $\mu$m. **d** Top: Gabor-filtered images of the filopodium from the yellow region in (**c**) at four consecutive time points (frame interval is 19 s) of a KP[fl]C cell in 4 mg/ml collagen I. Scale bar is 2.5 $\mu$m. Bottom: 3D tracks of the filopodium show bending and coiling. Red arrows indicate the binormal vectors along the filopodium. Brightness/contrast of the color channels (reflection and fluorescence) were adjusted individually.

uniformly expressed in the cells, also localized to the filopodia of the cells (Supplementary Fig. 10). Following depletion of mDia1 in MCF7 cells by siRNA (Supplementary Fig. 11) we did not observe a significant reduction in the angular velocity compared to MCF7 cells transfected with control siRNA (Supplementary Fig. 12).

Overall, these results strongly suggest that the molecular activity from myosin V and myosin X are somehow involved in the rotation of filopodia whereas the actin-binding protein mDia1 does not play a role in the observed rotations.

**Helical buckling and coiling of filopodia**. A clear indicator of axial rotation of actin in filopodia is the presence of helical buckling which arises from over-twisting of the actin shaft. In the following, we therefore focus on filopodia buckling in cells cultured in 3D collagen I networks in which dynamic filopodia can build up twist by interacting with the collagen I fibers. Filopodia from cells grown in 3D collagen I gels dynamically explore their 3D environment, but they are found to exhibit a more restricted motion compared to cells grown on glass due to the confinement of the fibers surrounding the cell, as shown in Fig. 4. Figure 4a–d shows a KP[fl]C cell labeled with mEmerald Lifeact-7, embedded in a 4 mg/ml collagen I gel, imaged with a confocal microscope.

Bending, buckling, and helical coiling of the filopodium became apparent as shown in Fig. 4 (see also Supplementary Movie 8).

The presence of an extracellular matrix could contribute to build-up of twist in the rotating filopodia. Specific and unspecific interactions between filopodia and the surrounding matrix will lead to twist in the rotating structure and further induce buckling in the case that sufficient twist is accumulated. Filopodia in cells migrating in a 3D collagen I network, are expected to experience external friction at the contact points between filopodium and the fibers[45]. Additionally, friction exists between the plasma membrane and the actin shaft mediated by various proteins linking the membrane with the actin. The helical buckles and coils observed in e.g. Fig. 4a–d, and Supplementary Fig. 1, 2, and Movie 8 can hence be a signature of over-twisting of the actin shaft which occurs naturally when rotating filopodia interact with collagen I network or the membrane.

**Filopodia buckle and pull at the same time**. To further investigate the nature of these buckles we tested whether such a twist-buckling transition occurs in presence of traction in the filopodium. We therefore measured the traction force when holding the membrane tether by an optical trap, as shown schematically in Fig. 5a. Compressive buckling from membrane tension, as suggested in refs. [27,46] is an unlikely mechanism if the filopodium is

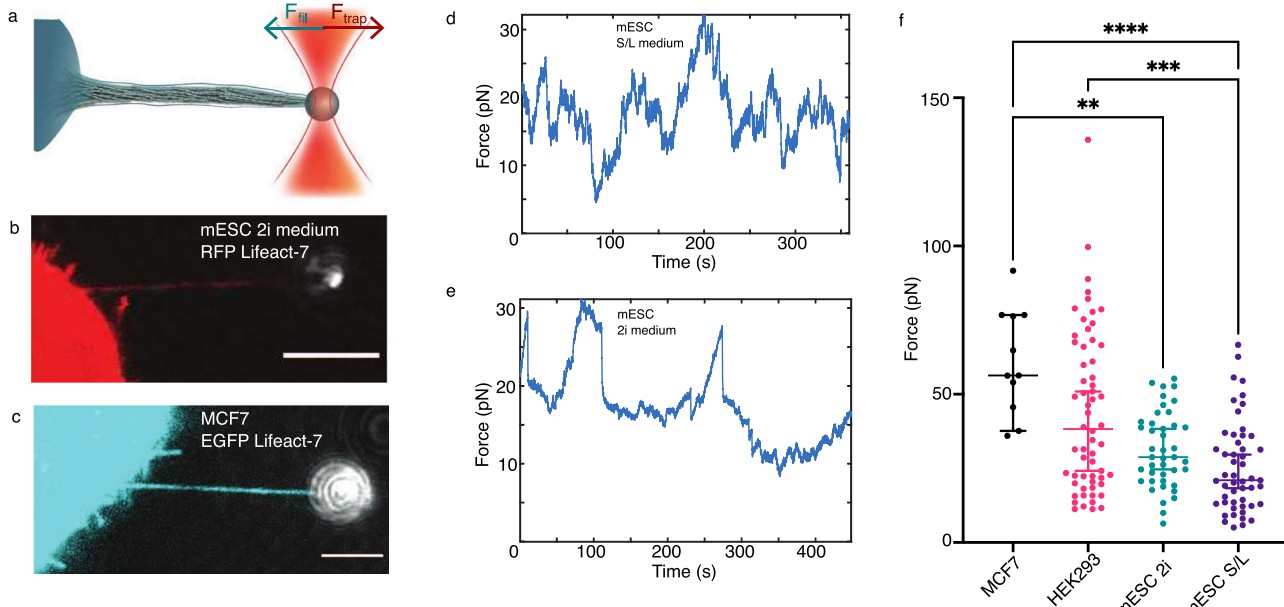

**Fig. 5 Tethers pulled from different cell types—from naive pluripotent stem cells to terminally differentiated cells—are highly dynamic and generate significant traction forces. a** Schematics of the optical tweezers assay to measure the dynamic force exerted by a tether. An optically trapped VN coated bead ($d = 4.95 \ \mu$m) is used to extract and hold a membrane tether at a given length with a holding force $F_{trap}$. **b** Representative confocal image of a filopodium extracted from a HV.5.1 mESC (red, RFP Lifeact-7) grown in 2i medium. Scale bar is 5 $\mu$m. **c** Representative confocal image of a filopodium extracted from a MCF7 cell (cyan, EGFP Lifeact-7). Scale bar is 5 $\mu$m. **d, e** Force curves for tethers pulled from HV.5.1 mESCs grown in 2i (**d**) and in S/L (**e**) medium. **f** Maximum tether holding forces measured for MCF7 ($N = 11$, mean ± std = (61.1 ± 17.9) pN), HEK293 ($N = 60$, (43.9 ± 27.1) pN), HV.5.1 mESCs cultured in 2i ($N = 42$, (31.3 ± 12.7) pN) and S/L medium ($N = 51$, (25.6 ± 15.4) pN). Kruskal–Wallis with adjusted $p$-value by Dunn's test with significance set at $p < 0.05$. Scatter plot shows the median and the whiskers extend from minimum to the maximum values. Adjusted $p$-values are 0.003 (MCF7/ mESC(2i)), <0.0001 (MCF7/mESC(S/L)), and 0.0004 (HEK293/mESC(S/L)). Source data are provided as a Source Data file. Brightness/contrast of the color channels (reflection and fluorescence) were adjusted individually.

able to exert a traction force while undergoing buckling. We observed buckling in filopodia, without the presence of external collagen I fibers, in optically trapped filopodia pulled from cells on glass, see Supplementary Fig. 3 and Supplementary Movie 9.

We extracted tethers from mouse embryonic stem cells and terminally differentiated cells (MCF7) that were transfected to express EGFP Lifeact-7 labeled F-actin, by optical trapping of a vitronectin coated bead ($d = 4.95 \ \mu$m) (Fig. 5b, c). After extracting a membrane tether with a trapped bead, small amounts of F-actin were immediately present inside the tether, but over a time course of ~150 s, the cell recruited more F-actin, as seen in Supplementary Fig. 13a, b for an HV.5.1 mESC, and the actin shaft was observed to extend several micrometers into the membrane tube. Subsequently, the new structure became highly dynamic and behaved like a filopodium (Supplementary Movies 10, 11). To exclude a contribution from the cell motility present in most cell types on the force, we performed parallel tracking of cells and force measurements. As seen in Supplementary Fig. 14 the cell movement is slower than the rapid changes in the force and are therefore uncoupled. The measured force can therefore be ascribed to tension in the filopodium.

To investigate the general traction force delivered by cells existing in different developmental stages we quantified the maximal filopodium traction forces exerted by MCF7, HEK293 cells, and HV.5.1 mESCs cultured in 2i or Serum/LIF (S/L) medium (Fig. 5d–f). We found that all cell types exclusively exert a traction force in the range 20-80 pN on the trapped bead and stem cells cultured in 2i or S/L medium were found to exert slightly lower maximal forces. Time traces of forces from MCF7 cells in Supplementary Fig. 15 show that the extracted tethers display similar activity as expected for filopodia. The high traction forces exceed the typical force of 10 pN needed for holding a pure

membrane tether containing no F-actin[4,18]. Since buckles can be observed in force generating tethers held by an optical trap we conclude that compressive forces from the membrane are unlikely to be responsible for the observed buckling and coiling of filopodia, but instead our data suggest that the actin structure exhibits an internal twist generating mechanism.

**Twist deformations are a generic non-equilibrium feature of confined actomyosin complexes.** Our experimental observations demonstrate that the complex dynamics of filopodia including helical rotation, buckling, and tip movement are induced by the twisting motion of actin filaments inside the filopodia. Furthermore, the emergence of such twisting motion in a variety of cell types and in both early and late stages of development points to a possible generic mechanism for the formation of twist in actin/ myosin complexes within the filopodial cell membrane. To understand the underlying mechanism of twist generation, we next used a three-dimensional active gel model to study the dynamics of actin/myosin complexes confined within a geometrical constraint. The existence of molecular activity in filopodia is well established through the presence of actin reorganization proteins[47–49] and molecular motors like myosin V and myosin X[6,37–40]. The choice of model was motivated by the generality of this class of continuum models which is due to the fact that only local conservation laws, interaction between the systems' constituents, and perpetual injection of energy at the smallest length-scale are assumed. Furthermore, active gel equations have proven successful in describing several aspects of the physics of actin/ myosin networks including actomyosin dynamics at the cell cortex[50,51], actomyosin induced cell motility[52,53], actin retrograde flows[54,55], and topological characteristics of actin filaments[56,57]. Within this framework the dynamics of actin/myosin complexes

are expressed through a continuum representation of their minimal degrees of freedom including the orientation and velocity fields. The orientation field is represented by a tensor order parameter $\mathbf{Q} = 3q/2 \times (\mathbf{nn}^T - \mathbf{I}/3)$, where $q$ is the magnitude of the orientational order and $\mathbf{n}$ is the director, representing the coarse-grained orientation of the actin filaments[58,59]. The dynamics of the orientation tensor $\mathbf{Q}$ follows Beris–Edwards equations[60], describing the alignment to flow and relaxation dynamics due to the filament elasticity $K$. The orientation dynamics is coupled to the velocity field governed by a generalized Stokes equation that accounts for the active stress generation due to force dipoles associated with actin tread-milling, described as $\mathbf{\Pi}^{\text{active}} = -\zeta \mathbf{Q}$, where $\zeta$ denotes the strengths of the active stress generation (see Materials and Methods for the details of the governing equations and the mapping of parameters to physical units).

In order to emulate the dynamics of actin filaments inside the filopodia, we consider a simplified setup of an active gel confined inside a three-dimensional channel with a square cross-section of size $h$. In two dimensions it is shown that increasing the confinement size above a threshold results in a spontaneous shear flow inside the confinement due to bend-splay deformations[61,62]. Interestingly, we find that here increasing the size of the three-dimensional confinement beyond threshold results in the spontaneous flow generation accompanied by the emergence of twist deformations (Fig. 6a, b). To characterize the amount of the twist in the system we measure the average twist deformations across the channel $\mathcal{T} = \langle |\mathbf{n} \cdot (\nabla \times \mathbf{n})| \rangle_{\mathbf{x},t}$, where $\langle \rangle_{\mathbf{x},t}$ denotes averaging over both space and time. As evident from Fig. 6c increasing the confinement size above a threshold results in increasing amount of twist in the active gel. The amount of twist generation and the threshold are further controlled by the activity of the filaments $\zeta$ and their orientational elasticity $K$. For a fixed channel size $h$ increasing activity triggers a hydrodynamic instability[61,63] that results in spontaneous flow generation and twist deformations (Fig. 6d) consistent with the experimental observations in Fig. 4 and Supplementary Fig. 16. The hydrodynamic instability and the creation of the spontaneous flows are suppressed by the filament elasticity $K$. As such increasing the filament elasticity reduces the amount of twist in the system up to a threshold value, where any twist generation is completely suppressed (Fig. 6e).

The interplay between the activity, elasticity, and the confinement size can be best understood in terms of a dimensionless parameter $\mathcal{A} = h \times \sqrt{\zeta/K}$ which describes the competing effect of two length scales: the confinement size $h$ and the length-scale set by combined effects of activity and elasticity $\sqrt{K/\zeta}$. Indeed plotting the amount of twist as a function of the dimensionless number $\mathcal{A}$ results in the collapse of the data corresponding to varying activity, elasticity, and confinement size (Fig. 6f), indicating that $\sqrt{K/\zeta}$ is the relevant activity-induced length scale: increasing the activity enhances active stress generation and orientational deformations, while such deformations are accommodated by the elasticity. As a result, larger activity (or smaller elasticity) leads to the emergence of deformations with smaller length scales. On the other hand, suppression of the activity is expected to increase the deformation length scale. When this length scale becomes larger than the confinement size, all deformations are suppressed and no twist is expected in the actin filaments.

**Chirality of twist**. It is important to note that the emergence of twist happens without having any explicit chirality in the equations of motion and is due to a hydrodynamic instability that is induced by the activity of the confined actomyosin complexes. Since this hydrodynamic instability breaks the mirror-symmetry, it will choose both clockwise or counterclockwise directions of rotation with a similar probability. From a biological perspective there could be several contributors for biasing the orientation of the twist. The molecular motors myosin V[25] and myosin X[64] have been shown to walk along a spiral path on actin bundles found in filopodia. These motors walk toward the tip in a counterclockwise orientation which would contribute to a torque in the opposite direction. Our experiments suggest that the rotation of filopodia is biased towards the opposite orientation of the path formed by these myosin motors which could suggest that motors increase the probability for the measured orientation. To capture this, experimentally measured bias, in our active gel dynamics we refine our model to include active stress originating from torque dipoles.

The effect of torque dipoles is accounted for through additional contributions to the active stress $\Pi_{\alpha\beta}^{\text{active torque}} = -\zeta' \epsilon_{\alpha\beta\gamma} \partial_\mu Q_{\gamma\mu}$, where $\epsilon_{ijk}$ is the Levi–Chevita operator and $\zeta'$ controls the strength of the torque dipole[50,65]. It is easy to see that this term already breaks the mirror-symmetry, such that positive (negative) $\zeta'$ exerts clockwise (counterclockwise) torque. No experimental measure of the relative strength of force and torque dipoles are available, however, dimensional analyses suggest that the ratio of active stress coefficients $\zeta'/\zeta \sim 0.1$ indicating that in general for actomyosin complexes the contributions of torque dipoles to the dynamics are small compared to the force dipoles (see Supplementary Information for the estimate of the torque dipole coefficient). Furthermore, the emergence of small, but finite number of clockwise rotations in the experiments, show that the hydrodynamic instability is the controlling mechanism for the mirror-symmetry breaking. Nevertheless, even small values of this chiral terms will bias the mirror-symmetry breaking in the direction of the torque dipole. Indeed our representative simulations show how changing the sign of $\zeta'$ results in the change in the handedness of the filaments rotation around the axis of the channel (see Supplementary Fig. 17).

## Discussion

Our results show that axial twisting and rotation are generic behaviors of cellular filopodia and was detected in naive pluripotent stem cells and in terminally differentiated cells. The spinning of the actin shaft leads to a rich variety of physical phenomena such as tip movement, helical buckling, and traction force generation and this allows the cell to explore the 3D extracellular environment while still being able to exert a pulling force as summarized in Fig. 7.

The rotational behavior of filopodia has been largely unexplored in literature. However, the periodic sweeping motion of macrophage filopodia was found to be 1.2 rad/s (0.191 Hz) as measured in two dimensions[22]. Growth cone filopodia were also tracked in two dimensions orthogonal to the axis of the filopodium and the frequency was found to be ~0.016 Hz[6]. In ref. [17] the actin structure of filopodia was imaged and small buckles were observed to rotate around the actin shaft, thus strongly indicating that actin has the ability to spin within the filopodium.

We explicitly performed 3D tracking of the whole filopodium and its tip, see Fig. 7c, d, e and made complimentary measurements of the spinning of the actin shaft within the filopodium, see Fig. 7f–h. Beads were added to the side of the filopodium by an optical trap and subsequently tracked in 3D. Vitronectin on the beads allows for binding directly to intergrin which can couple the bead to the internal actin structure. We note that beads can also bind non-specifically to plasma membranes[66] or to filopodia and still be coupled to the actin structure, as shown in ref. [67]. A clear signature that the bead is connected to the actin is

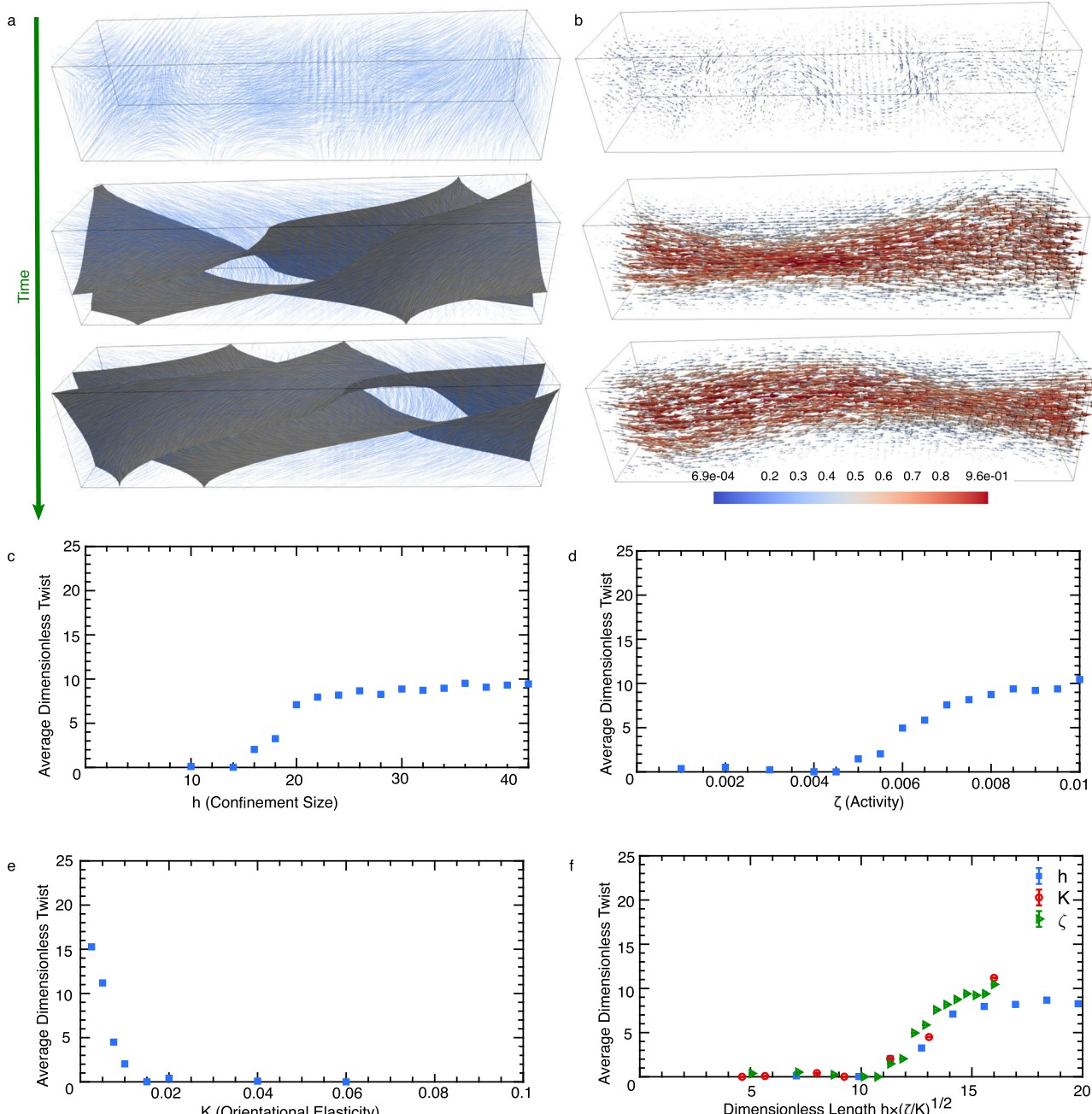

**Fig. 6 Actin shaft twisting is a generic phenomenon caused by the activity of actomyosin complexes inside the confining filopodial cell membrane. a, b** Temporal evolution of (**a**) the orientation and (**b**) velocity fields of the active gel representing actin filaments/myosin motor mixtures. In (**a**) the director field associated with the coarse-grained orientation of actin filaments $\vec{n}$ is shown by blue solid lines, and is overlaid with the isocontours of twist deformations $\vec{n} \cdot (\vec{\nabla} \times \vec{n})$. In (**b**) the colormap indicates the magnitude of the velocities normalized by the maximum velocity. **c–e** Dependence of the average twist on (**c**) the confinement size, (**d**) activity, and (**e**) elasticity. The average amount of twist is non-dimensionalized by the channel length. **f** Average twist as a function of the dimensionless length, for varying confinement sizes, activities, and elasticities, showing the collapse of the data. Source data are provided as a Source Data file.

translocation of the bead towards the cell body which is caused by retrograde flow within the filopodium. The results from these experiments showed that the actin shaft has the ability to spin with a similar frequency as the circular movement of the whole filopodium. These results strongly suggest that the spinning of the actin together with filopodia bending and growth, is responsible for the 3D motion and buckling of filopodia.

When a filopodium is free, we find that the tip rotates with a mean angular velocity of 0.4 ± 0.4 deg/s. In the presence of friction, arising from cells being embedded in a 3D matrix, we observe that rotation slows down and filopodia rotate with a mean angular velocity of 0.1 ± 0.1 deg/s. The external resistance to rotation experienced by filopodia leads to accumulation of twist and hence to helical buckling of the filopodium (Fig. 7c–e). This mechanism can be explained by twisting a rubber cable with one hand while holding the other end tight. The cable will accumulate twist, start to buckle and coil into an helical shape as shown in Supplementary Fig. 16. During buckling and coiling such a structure will shorten and therefore generate a pulling force.

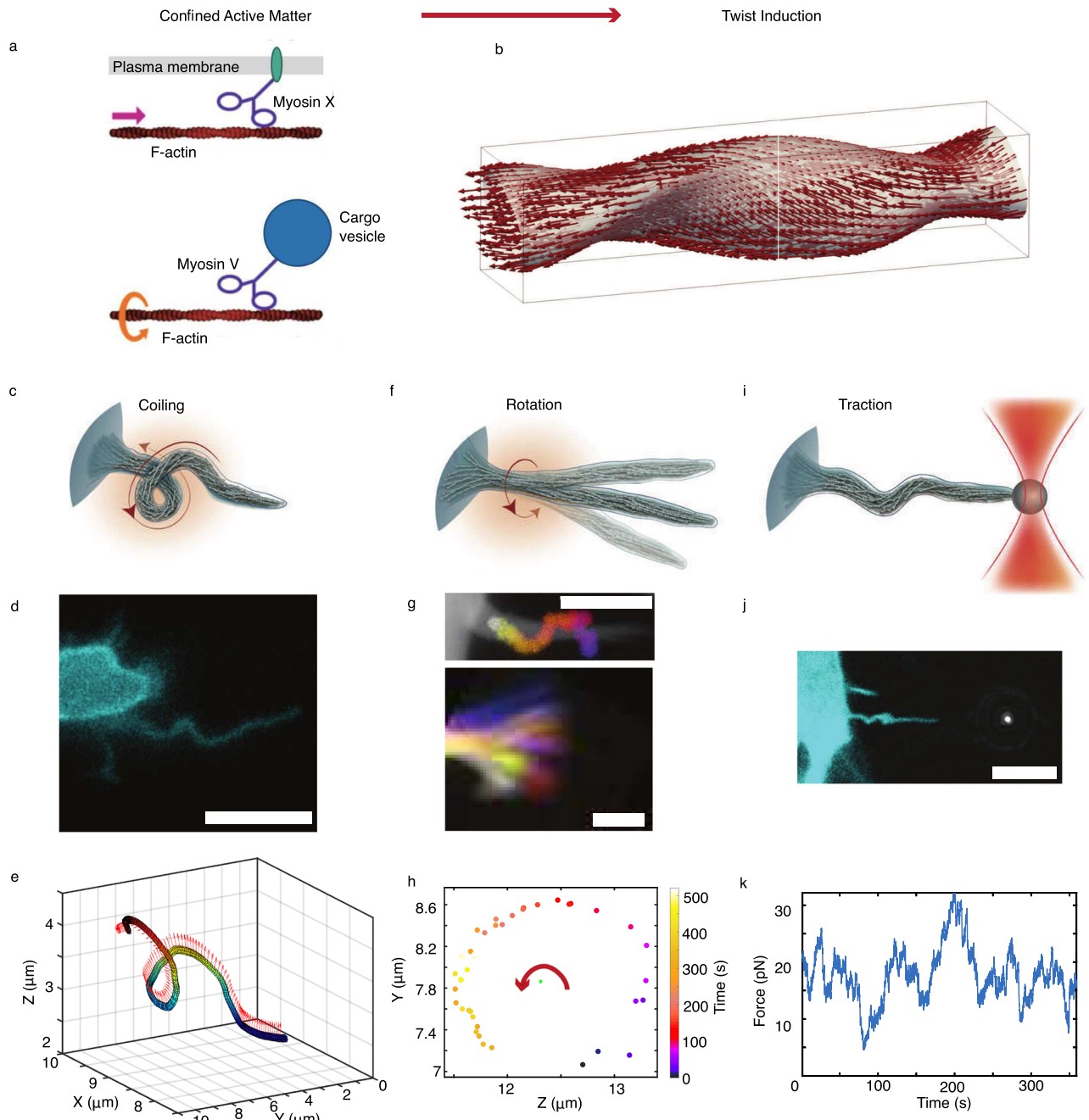

**Fig. 7 Twisting of the active filamentous actin core of filopodia is responsible for coiling, rotation, and traction by filopodia. a** Schematic of how walking of myosin motor on actin bundles, with inherent helical structures, generates force and torque dipoles. Myosin X and V walk in a helical motion around actin bundles and can make interfilament steps[25, 64]. The drag experienced from the viscous membrane (myosin X) and also from the attached cargo (myosin V) results in local forces applied to the actin structure. Similar molecular force dipoles originating from other myosins present in the cell cortex have been described previously[50]. **b** Filopodia twist is induced by active filaments under confinement of the filopodia membrane. **c–e** Schematic depiction of coiling (**c**), fluorescent image, scale bar is 5 $\mu$m (**d**), and 3D tracing of a filopodium from a MCF7 cell (cyan, EGFP Lifeact-7) (Supplementary Fig. 1) (**e**). Arrows in (**e**) indicate the binormals along the curve. **f** Schematic of a rotating filopodium. **g** Top: Confocal image of a filopodium extended from a MCF7 cell (white, EGFP Lifeact-7) overlayed with time color-coded images of the bead rotating around the actin shaft. Scale bar is 5 $\mu$m. A vitronectin-coated bead, indirectly attached to the actin fibers via transmembrane integrins, performs a spiral motion around the filopodium towards the cell body as a result of the internal twisting of actin shaft and retrograde actin flow (Fig. 2, and Supplementary Figs. 4, 5). Bottom: Gabor-filtered Z-projection of the movement of a filopodial tip of an EGFP Lifeact-7 labeled MCF7 cell at consecutive color-coded time points. Scale bar, 1 $\mu$m (Fig. 3). Actin shaft twisting and tip rotation occur at similar rates. **h** Tip tracking in the YZ plane reveals counterclockwise rotation as seen from the tip towards the cell body. **i** Schematic depiction of an optically trapped filopodium performing traction due to buckling and thus apparent shortening induced by accumulation of twist. **j** Buckling of a filopodium extracted from a HEK293T cell (cyan, EGFP Lifeact-7) using a 4.95 $\mu$m bead (gray, reflection). Scale bar is 5 $\mu$m. **k** Force profile from a filopodium extracted from a HV.5.1 mESC in S/L medium. Brightness/contrast of the color channels (reflection and fluorescence) were adjusted individually.

When actin is not specifically linked to an external structure, which can prevent spinning, buckling can still occur if friction exists between the membrane and the actin structure. In the presence of friction, for example from transmembrane proteins such as integrins and peripherally binding proteins like I-BARs[68], torsional twist can accumulate in the actin shaft and cause buckling[17,69]. Buckles form when twist is released and converted into bending energy within the actin shaft. This leads to an apparent shortening of the filopodium as observed when holding the filopodium by an optical tweezer[17] and thus contributes to a traction force in addition to the force arising from the retrograde flow of actin[18] (Fig. 7i–k).

We found the filopodia to predominantly rotate in a counterclockwise orientation, as seen from the tip towards the cell body. Of all resolvable rotations ($N = 68$, Fig. 3i, Supplementary Fig. 8 and Supplementary Table 2) we found 51% of the filopodia to rotate in a counterclockwise orientation while 9% rotated in a clockwise orientation as seen from the tip towards the cell. The orientation of 40% of the filopodia rotations could not clearly be resolved due to convoluted lateral movements, buckling, and rotations. Furthermore, in the control experiments in Fig. 3j and Supplementary Table 3 we observed a bias towards counterclockwise tip rotation with 49% exhibiting counterclockwise rotation, 15% rotated clockwise while 36% of rotation directions could not be resolved for a total of $N = 92$ filopodia. Interestingly, a similar sweeping motion was also measured for the movement of growth cone filopodia in ref. [6] which could indicate that filopodia are generically prone to exhibit counterclockwise rotation when observed from the perspective of the tip. However, to properly resolve the orientation of the spinning actin shaft it is necessary to perform experiments as in Fig. 2 where the actual spinning of the actin shaft can be isolated from the lateral movement of the filopodia.

Twist in actin bundles emerges naturally when active filaments are confined in channels with dimensions similar to a filopodia tube as shown by active matter simulations, (Fig. 6). Our model assumes active force generation along the actin shaft which could arise from the walking motion of myosin motor proteins along actin filaments. The presence of myosin V and myosin X in filopodia of cancer cells and other cell types is well established[6,37–40] and these motors can produce force and torque dipoles as shown schematically in Fig. 7a, b. Both of these motors experience a drag while walking along the actin bundle: myosin V from dragging vesicle encapsulated cargo while myosin X causes drag from transporting membrane-embedded proteins along the viscous plasma membrane as it walks towards the tip of the filopodium. Indeed we measured reduction in filopodia rotations upon mRNA silencing of myosin V or myosin X which supports the force generating ability of these motors on the actin structure. The induced chirality of the rotations is random, but can be biased by active torque dipoles in the system. Such dipoles indeed exist in natural filopodia in the form of myosins exhibiting chiral motion and interfilament stepping along actin bundles as demonstrated for the filopodia associated motors myosin V[25] and myosin X[64]. Both of these motors walk towards the tip in a filopodium and spiral around the actin in a clockwise orientation as seen from the tip thus inducing counterclockwise rotations, as seen from the perspective of the tip. Our experimental data indeed show a preference for counterclockwise orientation of the twist, see Supplementary Table 2, whereas mRNA silencing of myosin V and myosin X resulted in increased randomness for the orientation (Supplementary Table 3) thus supporting the idea that torque-generation by these motors induces a bias towards a specific chirality of the rotations. The presence of both twist orientations supports the theoretical predictions of a hydrodynamic instability being the origin of the twist.

To exclude possible artifacts arising from the transiently expressed fluorescent actin we also imaged cells which had the actin labeled with a membrane-permeable probe which binds to F-actin in living cells (SiR actin). These cells showed similar filopodial activity as we observed in cells transfected with EGFP Lifeact-7, see Supplementary Movies 6, 7.

The observation of similar filopodial dynamics in early stem cells confirms the generic nature of filopodial dynamics. The state of stem cells used here resemble the inner cell mass (ICM) cells, but their state can be controlled by different culture media. Cells cultured in serum-free medium supplemented with leukemia inhibitor factor (LIF) and small molecule inhibitors GSK3 and MEK (2i medium), genetically represent cells from the ICM of mouse blastocyst corresponding to 3.5 days post fertilization and are highly pluripotent[70,71]. Cells grown in Serum-LIF (S/L) medium are in a primed state towards epiblast[71]. The colonies of cells grown in these two media exhibit remarkably different morphologies; cells cultured in S/L medium are more spread out (Supplementary Fig. 13d) and form a monolayer of cells while 2i cultured cells grow together into an embryonic body (Supplementary Fig. 13c). The general role of filopodia in early development is less clear. Their presence has been detected in both the process of embryonic compaction, where they were shown to sustain tension, which was concluded from images, but the force was not quantified[9,72]. Our quantitative data show that naive pluripotent embryonic stem cells are able to deliver a traction force on the order of 10 pN for each filopodium. However, whether the spinning motion of actin does occur during compaction, where the filopodia are penetrating other cells, remains an open question. Filopodia in mesenchymal stem cells have been shown to function as rails facilitating transport of morphogens between cells[2]. Such tunneling tethers or cytonemes have been observed in many cell types and we have also detected rotation of such structures in Hepa 1–6 mouse hepatocytes (Supplementary Movies 6,7), however, the functional role of the rotation remains elusive in these structures.

Altogether, our experiments and theoretical modeling present evidence of a general rotary mechanism observed in filopodia. Remarkably, this rotary mechanism exists for a wide selection of different cell types ranging from naïve stem cells to terminally differentiated cancer cells, indicating that the observed phenomenon must be generic within various cell types. The twist generation in filopodia adds an additional way for cells to explore their 3D extracellular environment, to navigate through dense networks of the ECM and to perform chemical sensing. The spinning or twisting motion of actin observed in this work can well exist in other actin rich locations in cells, exhibiting molecular activity, and hence could have more general implications for cell function.

## Methods

**Cell culture**. All cells were grown in vented T25 flasks (BD Falcon) in a sterile environment at 37 °C in a humidified 5% $CO_2$ incubator. The cells were passaged at around 80% confluence.

HEK293 (ATCC #CRL-1573), HEK293T (ATCC #CRL-11268), and MCF7 (provided by Dr. David Springs, University of Wisconsin) cells were grown in DMEM growth medium ([+] 4.5 g/L D-Glucose, [+] L-Glutamine, [+] 110 mg/L Sodium Pyruvate, Gibco, supplemented with 10% FBS, 1% PenStrep) and passaged by washing with 2 ml DPBS (1X, [−] $CaCl_2$, [−] $MgCl_2$, Gibco or PBS pH 7.4, Gibco), detached using 1 ml TrypLE Express ([−] Phenol red, Gibco) and re-suspended in growth medium.

MCF7-p95ErbB2 cells described previously[35] were grown in DMEM growth medium ([+] 4.5 g/L D-Glucose, [+] L-Glutamine, [+] 110 mg/L Sodium Pyruvate, Gibco, supplemented with 10% FBS, 1% PenStrep, further supplemented with 200 μg/ml G418, 1 μg/ml Puromycin, 0.5 μg/ml tetracycline). To induce p95ErbB2 expression, the cells were trypsinized and washed 4–5 times in 20 ml of PBS (+$CaCl_2$) to remove all tetracycline and then plated in a new tissue culture flask. When cells were induced, they were keep in DMEM growth medium ([+] 4.5 g/L D-Glucose, [+] L-Glutamine, [+] 110 mg/L Sodium Pyruvate, Gibco,

supplemented with 10% FBS, 1% PenStrep). The "spiderlike" phenotype of the MCF7-p95ErbB2 cells appears after 3–5 days.

Uninduced MCF7-p95ErbB2 cells, meaning cells grown in growth medium containing tetracyline, will express some ErbB2 as the promotor is leaky, but will resemble more normal MCF7 cells in their activity levels.

KP^{R172H}C (KPC), KP^{fl}C cells were cultured in DMEM containing GlutaMAX, 10% FBS, and 1% PenStrep. Cells were a kind gift from Jennifer Morton (Beatson Institute).

Mouse embryonic stem cells (HV.5.1 mESCs) were obtained from the Brickman Group (DanStem, University of Copenhagen, Denmark). HV.5.1 cells are equipped with a Venus reporter which represents the expression state of Hex, line. HV.5.1 mESCs had been passaged 23 times prior to our experiments. The cultures were sustained for no longer than 25 subsequent passages. Stem cells were either cultured in Serum/LIF (S/L) or 2i culture medium

Serum/LIF (S/L) medium. The addition of the cytokine Leukemia Inhibitory Factor (LIF) maintains a pluripotent state of the stem cells[73–75]. Cells were cultured in Glasgow modified Eagle's medium (GMEM, Sigma Aldrich, Germany) supplied with 5.5 ml non-essential amino acids (Gibco Science, Paisley, UK), 5.5 ml glutamine, 5.5 ml sodium pyruvate (Gibco Science, Paisley, UK), 560 $\mu$l 0.1 mM 2-merceaptoethanol (Sigma-Aldrich, St. Louis, MO, USA), 50 ml fetal bovine serum (FBS) (obtained from DanStem, University of Copenhagen, Denmark), 560 $\mu$l 1000 U/ml LIF (obtained from DanStem, Copenhagen University, Denmark) per 500 ml of medium. The entire pile of FBS was bought from the manufacturer by DanStem to ensure consistency in the culture medium. For S/L culture, the medium was aspirated, and rinsed once with DPBS and detached with 0.025% trypsin in DPBS, incubated for 4 min at 37 °C and suspended in 5 ml of S/L medium. The cells were then centrifuged and re-suspended in 5 ml of S/L.

2i medium. In 2i medium, cells remain in a more primitive state. 2i composition: 1:1 mixture of DMEM (Gibco Science, Paisly, UK) and F12 medium, modified N2 medium (25 $\mu$g/ml insulin, 100 $\mu$g/ml apo-transferrin, 6 ng/ml progesterone, 16 $\mu$g/ml putrescine, 30 nM sodium selenite, obtained from DanStem, University of Copenhagen, Demark), 50 $\mu$g/ml bovine serum albumin fraction V, combined with a 1:1 mixture of Neurobasal medium supplemented with B27 (Gibco, Paisley, UK). The culture medium contained two cytokines, a GSK3 inhibitor with a final concentration of 1 $\mu$M and ERK inhibitor with a final concentration of 3 $\mu$M. The cells were cultured in 50 ml Falcon flasks (Falcon Science, Paisley, UK) coated with EmbryoMax ultrapure water with 0.1% gelatin (Merck Millipore, Billerica, MA, USA) for 10 minutes. At a confluence of approximately 90% the cells were harvested. Cells detach from the culture flask with time. To retain a large number of cells, the culture medium was collected in a 15 ml Falcon tube and centrifuged for 3 min at a speed of 1890 rpm prior to trypsinization. The supernatant was aspirated and the pellet was re-suspended in 1 mL of 0.025% trypsin by pipetting. The suspension was added to the flask in which the cells had been cultured to detach the clusters of cells that had remained bound to the substrate and incubated at 37 °C for 4 min. Cells were transferred to a 15 ml Falcon tube and 5 ml of N2B27 was added to reduce the usage of the cytokines and LIF. The suspension was centrifuged with the same settings and re-suspended in 5 ml of N2B27 medium, by pipetting carefully up and down. Finally, a volume of the cell suspension was added to a new culture flask together with 5 ml of fresh 2i medium.

**F-actin labeling**. Transient transfection of HEK293, HEK293T, and MCF7 cells. The day prior to transfection, the cells were seeded on a 24 well cell culture plate (Orange Scientific). On day of transfection, the medium in each well was removed and exchanged with 500 $\mu$l fresh growth medium per well. Then 1 $\mu$l of EGFP Lifeact-7 plasmid (mEGFP Lifeact-7 was a gift from Michael Davidson; Addgene plasmid # 54610), 100 $\mu$l Opti-MEM ([+] HEPES, [+] 2.4 g/L Sodium Bicarbonate, [+] L-Glutamine, Gibco), and 1 $\mu$l Lipofectamine LTX Reagent (Invitrogen) were carefully mixed in an Eppendorf tube and incubated at RT for at least 30 min. This mix was sufficient for a single well. Usually, the cells in at least 3 wells were transfected at a time. The next day, the cells were detached from the wells as described above, the contents of the 3 wells were mixed together. 100 $\mu$l of those cells and ca 1 ml of growth medium was added to a glass bottom dish (35 mm, No. 1.5 coverslip, 20 mm glass diameter, uncoated, MatTek) and left to adhere and express for another day in the incubator. The samples for experiments were then used 1 or 2 days after transfection.

Transient transfection of KPC and KP^{fl}C. KPC and KP^{fl}C cells were transfected with mEmerald Lifeact-7 (a kind gift from the Ivaska lab, Turku Centre for Biotechnology) using Lipofectamine 2000 as described above. After an incubation period of ~12 h they were embedded in collagen I matrices.

Transient transfection of HV.5.1 mESCs. Cells were transfected with EGFP Lifeact-7 or RFP Lifeact-7 using Lipofectamine 3000 in 24 well plates. The day before transfection, the cells were plated at high density (70−90% confluency). On the day of transfection, a transfection mixture containing Opti-MEM, plasmid, Lipofectamine 3000, Lipofectamine 3000 reagent, and medium was mixed and incubated for 12 minutes at room temperature. The medium was aspirated from the cells and the mix was subsequently added and was equally distributed over the well. Thereafter the well was filled with fresh medium. Although the manufacturer recommends to use Opti-MEM medium for better transfection efficiency, the cells were cultured in S/L or 2i medium, respectively, to maintain an optimal pluri- or

multipotent state. The cells were used for experiments between 24 and 48 hours after transfection.

SiR actin labeling. Hepa 1–6 cells were incubated in SiR actin (SiR actin kit, Spirochrome, 1: 2000) and Verapamil (1:1000) for 5 h.

**Collagen I gels**. Collagen mixtures of 4 mg/ml were prepared by mixing high concentration acid-extracted and cross-linked rat tail collagen I, sterile phosphate-buffered saline (PBS), and 5X collagen buffer containing 0.1 M HEPES, 2% NaHCO$_3$, and α-MEM. Cells were then suspended in the collagen mixtures. After polymerization, the gels were washed once and incubated with normal culture medium for 24 h.

**Vitronectin coated beads**. We prepared two stock solutions: 'tracer beads' with $d = 0.99$ $\mu$m streptavidin coated Flash Red fluorescent beads (Bangs Laboratories) and 'tether beads' with 4.95 $\mu$m streptavidin coated polymer particles (Bangs Laboratories) coated with biotin labeled human multimeric vitronectin (VN) (Innovative Research) in PBS (pH 7.4, Gibco) which were stored at 4 °C. For coating, 5 $\mu$l of beads were washed twice in 1 ml of PBS. After the second washing step, the beads were re-suspended in 50 $\mu$l of PBS containing 2 $\mu$l of VN, enough to exceed the maximal binding affinity of the streptavidin coated beads. The beads were incubated for 20 min at a shaking platform (Eppendorf MixMate, Hamburg, Germany) with a speed of 450 rpm. To remove unbound VN, the beads were centrifuged at 12,000 rpm for 5 min and re-suspended in 1 ml of PBS.

**Imaging sample preparation**. On day of experiments, 400 $\mu$l of 'imaging DMEM' ([+] 4.5 g/L D-Glucose, [+] L-Glutamine, [+] 25 mM HEPES, [−] Sodium Pyruvate, Gibco) was added to an Eppendorf tube together with 3 $\mu$l and 7 $\mu$l of 1 $\mu$m and 4.95 $\mu$m VN coated beads, respectively. The medium in a MatTek dish with transfected cells (see section 'Transient transfection') was carefully removed and replaced with the described bead mix in imaging DMEM.

**Silencing control experiments**. siRNAs and transfection. Myosin Va, Vb, X and mDia1 siRNAs were purchased from Sigma Aldrich and control siRNA (CSI) were purchased from Qiagen. The siRNA efficiency was validated with Western blots, see Supplementary Fig. 9 and Supplementary Fig. 11.

Reverse siRNA transfections of MCF7 cells were performed using Oligofectamine transfection reagent (Invitrogen) with 25 nM siRNA (Sigma-Aldrich) according to the manufacturer's protocol. The siRNA containing medium was replaced after 24 h and the analysis performed after 72 h.

SiRNAs were purchased from Sigma Aldrich unless otherwise stated. SiRNA sequences:

Control siRNA (AllStars Negative Control siRNA, 1027281 Qiagen)
MYO5A#1 siRNA (5′-CUGACUACCUGAAUGAUGA-3′),
MYO5A#2 siRNA (5′-CGAAACAACUGGAACUCGA-3′),
MYO5B#1 siRNA (5′-GACAUAGAUUUGGACCCGA-3′),
MYO5B#2 siRNA (5′-GAGAUCAUCCUGCAGGUAU -3′),
MYO10#1 siRNA (5′-CUUACGAAUCUCUUAAGAA-3′),
MYO10#2 siRNA (5′-GAAUCAGUCUGGAUGUGUA-3′),
mDia1#1 siRNA (5′-CAUGUGAGGAGUUACGUAA-3′),
mDia1#2 siRNA (5′-GACAGAAGAAGGAAUCCUA-3′).

For live cell control experiments, siRNA transfections were performed using Oligofectamine transfection reagent and cells were kept in a humidified incubator at 37 °C and 5% CO$_2$. The culture medium was renewed after 24 h.

Plasmid transfection. 48 h post-RNAi, the cells were transiently transfected with the plasmid(s) of interest (Lifeact GFP plasmid, Lifeact-mCherry plasmid, a kind gift from Roland Wedlich Söldner; mEmerald-mDia1-C-14, a kind gift from Michael Davidson, Addgene plasmid # 54156) using Lipofectamine LTX according to the manufacturer's protocol. Briefly, 1.25 $\mu$g plasmid was diluted in 500 $\mu$l OptiMem and 5 $\mu$l Lipofectamine LTX was added at RT for 25 min. After washing the cells with PBS the transfection mixture was added. The volume was adjusted with OptiMem and cells were kept in the humidified incubator at 37 °C and 5% CO$_2$. After 2 h 45 min, the medium was replaced with DMEM supplemented with 10% FBS and 1% PenStrep.

Immunoblotting. Cells were lysed in Laemmli sample buffer (125 mM Tris, pH 6.7, 140 mM SDS, 20% glycerol, 0.3 $\mu$M bromophenol blue) supplemented with protease inhibitor cocktail (Roche 4693124001), phosphatase inhibitor (Roche 4906837001) and 0.1 M dithiothreitol (DTT). Cell lysates were boiled for 5 min and separated by SDS-PAGE using precast 4–15% gradient gels (BioRad) followed by transfer to nitrocellulose membranes (BioRad) using Trans-Blot Turbo™ transfer system and blocked in PBS, containing 0.1% Tween-20 (PBST) and 5% BSA. The molecular weights of proteins from the gels were evaluated using Novex™ Sharp Protein Standard (Invitrogen). Membranes were incubated with primary antibodies in PBST/5%BSA at 4 °C overnight (mDia1 1:500 dilution, BD Transduction Laboratories 610848; Myosin 5a, 1:1000 dilution, Cell Signaling Technology 3402; Myosin 5b, 1:500 dilution, Novus Biologicals NBP1-87746; Myosin 10, 1 $\mu$g/mL, Sigma Aldrich HPA024223; Hsp90, BD Transduction Laboratories 610418, 1:4000, GAPDH, 1:7500 dilution, Abcam ab189095). Membrane was washed followed by incubation with appropriate peroxidase-conjugated secondary antibodies (anti-rabbit immunoglobulin G (IgG), Vector Laboratories, PI-1000; anti-mouse IgG,

Dako, P0260; both 1:5000) for 0.5 h at RT. Chemiluminescent signals (Clarity Western ECL substrate, BioRad) were detected with Luminescent Image Reader (LAS-1000Plus, Fujifilm).

Sample preparation for imaging. On the day of imaging, the transfected cells were detached from the 6-well plate by pipetting and part of the cell suspension transferred to a Mattek glass bottom dish were incubated for 5 h to have cells attached to the glass surface of the dish. Prior to imaging, the culture medium was replaced with FluoroBrite DMEM.

**Experimental setup and procedures.** Confocal microscopes with integrated optical trap. Experiments were conducted on two different setups. One setup consists of an inverted Leica TCS SP5 II confocal microscope with a 63X water objective (1.2 NA, Leica) with an integrated optical trap based on a 1064 nm laser (Nd:YVO4, 5 W Spectra Physics BL106C, $\lambda = 1064$ nm, TEM∞) and a photodiode detection system. The laser beam was tightly focused by the water-immersion objective and the trapping laser light was collected by a condenser (Leica, P1 1.40 oil S1) located in the back-focal plane and focused onto a quadrant photodiode (S5981; Hamamatsu). A three-dimensional LabVIEW controlled piezoelectric stage (Nano-Drive, MCL) allowed positioning of the sample relative to the laser focus with nanometer precision. For some experiments, the sample stage was heated to 39 °C using a stage heater which, due to thermal losses, results in a sample temperature of about 37 °C inside the sample. Data were acquired by an acquisition card (NI PCI- 6040E) at a sampling frequency of 1 kHz (for force measurements) and 22 kHz (for force calibration) and processed by custom-written LabVIEW programs (LabVIEW 2010; National Instruments). The second setup, mostly used for force measurements and stem cell experiments, is described in[76].

Free filopodia and filopodia in collagen rotations. Cells in MatTek dishes containing imaging medium or grown in collagen I gels were placed on the confocal microscope and filopodia movement was imaged via Z-stacks over time. The mEmeraldLifeact-7 or EGFP Lifeact-7 expressing cells were excited at 488 nm and the collagen I gel was imaged in reflection mode using the 633 nm laser without enhanced dynamics.

Bead rotations on tether. For experiments where the movement of a VN coated bead along a tether was followed, a MatTek dish with transfected cells and VN coated beads was placed on the sample stage. In the following, we distinguish two types of VN coated beads: 'tether beads' with $d = 4.95$ μm, and 'tracer beads', with $d = 0.99$ μm, see section 1.4 for VN coating details. During tether extraction we imaged the sample in bright field mode to avoid confocal laser radiation before the actual data acquisition. A tether bead was optically trapped and brought close to a (usually isolated and spread out) cell. The bead was carefully pressed against the cell membrane to allow attachment and then slowly moved a distance of about 10 μm (using the LabVIEW controlled piezo stage) such that a membrane tether was extracted between cell and bead. Once a tether was extracted, we switched to confocal mode. The EGFP Lifeact-7 was excited at 488 nm, the flash red tracer beads at 633 nm. The tether bead was carefully pushed onto the sample glass surface such that it got stuck and the trapping laser could be turned off. A control Z-stack was acquired (the tether beads were imaged in reflection mode using the 633 nm laser without enhanced dynamics) to ensure the tether was held and did not stick on the glass surface. Then a tracer bead was caught using the optical trap and carefully attached onto the tether close to the tether bead. Confocal Z-stacks over time were acquired to follow the tracer bead's rotation around the tether towards the cell body.

**Data analysis and image acquisition.** Image acquisition was done using Leica Application Suite (LAS) and processing and analysis were done using ImageJ and custom-written MATLAB scripts.

Tracking the movement of a bead attached to a filopodium. To localize the position of an attached bead on a filopodium, the XY position of the center of the bead and the filopodium were extracted from the Z-projection of the image stack, see Supplementary Fig. 6. Afterward, the Z-positions in each time point were extracted from the orthogonal view of the stack, shown in Supplementary Fig. 6b, which corresponds to the red dashed line in Supplementary Fig. 6a.

Tracking the rotation of filopodial tips. To quantify the tip rotation of filopodia, we increased the resolution of the images using a Gabor filter[77,78]. Then the center of a filopodium was extracted by fitting Gaussian function to the orthogonal view along the filopodium, see Supplementary Fig. 7.

Tracking helical filopodial coils. We used a 3D segmentation plugin software (Simple Neurite Tracer) in ImageJ[79] to track filopodia after removing the background intensity in images using Gabor and Gaussian blur filters (Supplementary Fig. 8).

**Statistical analysis.** Statistical analysis was performed using Prism software (GraphPad, version 9.2.0). p-values were obtained using the appropriate tests (described in the individual Fig. captions) with significance set at $p < 0.05$. Graphs show symbols describing p-values: $*p < 0.05$; $**p < 0.01$; $***p < 0.001$, $****p < 0.0001$. Unless otherwise stated, the box plots show the median and the whiskers extend from the maximum to the minimum values. Figure 1a: Representative confocal image, filopodial tips of HEK293T cells were tracked in >15 independent samples. Figure 1c (and Supplementary Fig. 2a): Representative

confocal image, filopodial buckles of KPC cells embedded in 4mg/ml collagen I, buckles were observed in 3 independent samples. Figure 1e: Tether pulling experiments with confocal imaging and simultaneous force measurements on MCF7 cells were conducted in >12 independent samples. Figure 5b: 42 tethers (from mESC cultured in 2i medium) and 51 tethers (from mESC cultured in S/L medium) were extracted during confocal imaging with simultaneous force measurements. Figure 5c: The same type of experiments as described above for Fig. 1e. Figure 7d: Coiling of MCF7 filopodia was observed in 8 independent samples. Figure 7g: (same data as shown in Supplementary Fig. 3a): Buckles during tether pulling experiments with confocal imaging and simultaneous force measurements on MCF7 cells were observed in >10 independent samples.

**Simulation methods.** We use the 2D equations of active nematohydrodynamics based on the theory of liquid crystals, which have proven successful in describing spatio-temporal dynamics of the cell cytoskleton, including actomyosin complexes[54–57] and microtuble bundles powered by kinesin motor proteins[80–82]. The orientational order of microscopic actin filaments is represented by the nematic tensor $\boldsymbol{Q} = \frac{3q}{2}(\mathbf{nn}^T - \boldsymbol{I}/3)$, with q the magnitude of the orientational order, $\mathbf{n}$ the director, and $\boldsymbol{I}$ the identity tensor, which evolves as

$$(\partial_t + \mathbf{u} \cdot \nabla)\boldsymbol{Q} - \boldsymbol{S} = \Gamma \boldsymbol{H}, \tag{1}$$

where $\Gamma$ is a rotational diffusivity and the co-rotation term

$$\boldsymbol{S} = (\lambda \boldsymbol{E} + \boldsymbol{\Omega}) \cdot \left(\boldsymbol{Q} + \frac{\boldsymbol{I}}{3}\right) + \left(\boldsymbol{Q} + \frac{\boldsymbol{I}}{3}\right) \cdot (\lambda \boldsymbol{E} - \boldsymbol{\Omega})$$
$$- 2\lambda \left(\boldsymbol{Q} + \frac{\boldsymbol{I}}{3}\right)(\boldsymbol{Q} : \nabla \mathbf{u}), \tag{2}$$

accounts for the response of the orientation field to the extensional and rotational components of the velocity gradients characterized by the strain rate $\boldsymbol{E} = ((\nabla \mathbf{u})^T + \nabla \mathbf{u})/2$ and vorticity $\boldsymbol{\Omega} = ((\nabla \mathbf{u})^T - \nabla \mathbf{u})/2$ tensors. The relative strength of extensional and rotational flows is determined by the alignment parameter $\lambda$. Therefore, the alignment parameter accounts for the different responses of particles of different shapes to the symmetric and asymmetric parts of the velocity gradient tensor[83]. Mapping the alignment parameter to the Leslie-Ericksen equation for liquid crystal dynamics gives $\lambda = \frac{3q+4}{9q} \frac{\beta^2 - 1}{\beta^2 + 1}$[59], where $\beta = a/b$ is the ratio of the length of the cell along its axis of symmetry, a, to its length perpendicular to this axis, b. Therefore, for prolate ellipsoids $\beta > 1$, while for oblate ellipsoids $\beta < 1$ and for spherical particles $\beta = 1$, which correspond to $\lambda > 0$, $\lambda < 0$, and $\lambda = 0$, respectively. Actin filaments are rod-shaped elongated particles and therefore are characterized by $\lambda > 0$. The relaxation of the orientational order is determined by the molecular field,

$$\boldsymbol{H} = -\frac{\delta \mathcal{F}}{\delta \boldsymbol{Q}} + \frac{\boldsymbol{I}}{3} \text{Tr}\left(\frac{\delta \mathcal{F}}{\delta \boldsymbol{Q}}\right), \tag{3}$$

where $\mathcal{F} = \mathcal{F}_b + \mathcal{F}_{el}$ denotes the free energy. We use the Landau–de Gennes bulk free energy for $\mathcal{F}_b$[83],

$$\mathcal{F}_b = \frac{A}{2}\boldsymbol{Q}^2 + \frac{B}{3}\boldsymbol{Q}^3 + \frac{C}{4}\boldsymbol{Q}^4, \tag{4}$$

and $\mathcal{F}_{el} = \frac{K}{2}(\nabla \boldsymbol{Q})^2$ describes the cost of spatial inhomogeneities in the order parameter, assuming a single elastic constant K. There are in general three elastic constants, corresponding to bend $K_b$, splay $K_s$, and twist $K_t$ in 3D systems. However, setting different values of $K_b$, $K_s$, and $K_t$ does not change the mechanism of the hydrodynamic instability.

The velocity field is evolved according to the incompressible Navier–Stokes equation:

$$\nabla \cdot \mathbf{u} = 0, \tag{5}$$

$$\partial_t \mathbf{u} + \mathbf{u} \cdot \nabla \mathbf{u} = \nabla \cdot \boldsymbol{\Pi}, \tag{6}$$

which reduces to the force balance equation $\nabla \cdot \boldsymbol{\Pi} = \mathbf{0}$ in the low Reynolds number limit relevant to actin filaments mechanics, with a stress tensor $\boldsymbol{\Pi}$ that must account for contributions from the viscous stress $\boldsymbol{\Pi}^{visc} = 2\eta \boldsymbol{E}$ and the elastic stress

$$\boldsymbol{\Pi}^{elastic} = -P\boldsymbol{I} + 2\lambda(\boldsymbol{Q} + \boldsymbol{I}/3)(\boldsymbol{Q} : \boldsymbol{H})$$
$$- \lambda \boldsymbol{H} \cdot \left(\boldsymbol{Q} + \frac{\boldsymbol{I}}{3}\right) - \lambda \left(\boldsymbol{Q} + \frac{\boldsymbol{I}}{3}\right) \cdot \boldsymbol{H} \tag{7}$$
$$- \nabla \boldsymbol{Q} : \frac{\delta \mathcal{F}}{\delta \nabla \boldsymbol{Q}} + \boldsymbol{Q} \cdot \boldsymbol{H} - \boldsymbol{H} \cdot \boldsymbol{Q},$$

which includes the pressure $P$[60]. The active contribution to the stress is accounted for by $\boldsymbol{\Pi}^{act} = -\zeta \boldsymbol{Q}$[63], such that any gradient in $\boldsymbol{Q}$ generates a flow field, with strength determined by the activity coefficient, $\zeta$. This active stress term accounts for the local stresses generated by active processes in the cells, including actomyosin polymerization and contractility[54,55].

The equations of active nematohydrodynamic are solved using a hybrid lattice Boltzmann and finite-difference method[84] that does not include thermal fluctuations. The momentum equation is solved using the lattice Boltzmann

method to resolve the hydrodynamics, and the method of lines is implemented to determine the order paramter in Eq. (1). A finite-difference approach is used for spatial discretization of Eq. (1) and the temporal evolution is obtained through an Euler integration scheme.

A three-dimensional rectangular domain with a square cross-section is used for simulations. The length ($L$) of the channel was fixed as 128 while the height ($h$) was varied between 4 to 32. In our study, simulations are initialized with a stagnant fluid and randomly oriented director field. Zero velocity and no flux of $Q$ at the walls are used as the boundary conditions, therefore no anchoring boundary condition is imposed for the orientational order parameter on the channel walls.

The parameters used in the simulations are $A = 1.0$, $\Gamma = 0.34$, $\eta = 2/3$, and $\lambda = 0.7$, in lattice units. As usual in lattice Boltzmann schemes, discrete space and time steps are chosen as unity and all quantities can be converted to physical units in a material-dependent manner[84–86]. In order to map the simulation parameters to physical units we consider the typical length of the filopodia as the characteristic length $L_0 \sim 10\,\mu m$, and the experimentally measured viscosity and elasticity of the actin/myosin mixtures[56,57] as the characteristic viscosity $\eta_0 = 0.1\text{Pa}.s$ and force units $F_0 = 10$ pN, respectively. This maps the confinement sizes studies here to the range $\sim 0.2 - 2\,\mu m$, elasticities to the range $\sim 0.1 - 1.0$ pN, and the activities to $\sim 0.01 - 0.1$Pa, consistent with the values reported for in vitro two-dimensional extract of actin/myosin mixtures[56].

Furthermore, the magnitude of the torque dipole can be estimated based on analytical derivation of[65], which suggests:

$$Ma = \frac{2\pi far^2}{(4\pi^2 r^2 + P^2)^{1/2}},\qquad(8)$$

where $M$ is the torque monopole, $a \sim 100$ nm distance between filaments, $f \sim 10$ pN is the force generated by myosin motors, $r \sim 5$ nm the radius of actin filament, and $P \sim 72$ nm is the helical pitch of the actin filaments. Using these values the ratio of the active stress associated with the torque dipole $\zeta'$ to the active stress due to force dipole $\zeta$ can be estimated as $\zeta'/\zeta \sim M/fa \sim 0.1$, which is the value mentioned in the main text. Therefore, the mechanism of symmetry-breaking and twist generation is governed by hydrodynamic instabilities that are induced by the active stress associated with force dipoles. The role of torque dipole is to set the direction of the twist: in the absence of a torque dipole $\zeta' = 0$ (results in the main text) clockwise and counterclockwise rotations are equally likely. Including a torque dipole is expected to bias this twist towards clockwise and counterclockwise rotation depending on the sign of $\zeta'$. To test this, we performed simulations with $\zeta' = (0, 0.1 \times \zeta, \zeta)$ and compared the distribution of clockwise and counterclockwise rotations for each case—with random initial conditions. As evident from Supplementary Fig. 9 increasing the value of $\zeta'$ clearly increases the biased rotation and when it becomes comparable to the value of active stress associated with force dipoles it can deterministically set the direction of rotation.

**Reporting summary**. Further information on research design is available in the Nature Research Reporting Summary linked to this article.

## Data availability
The data generated in this study are available from the corresponding author upon request. Source data are provided with this paper and example raw data sets are provided together with the available code (see Code availability). Source data are provided with this paper.

## Code availability
The Matlab codes used for analyzing the data in this work are available through github: [https://github.com/Younes-FB/Gabor_image_filtering] (Image filtering). [https://github.com/Younes-FB/filament_3Dtrack] (Code for 3D tracking of filopodia). [https://github.com/Younes-FB/Track_bead_On_tube_3D] (Code for tracking beads on filopodia).

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

## Acknowledgements

P.M.B. and N.L. acknowledge support from the Danish Council for Independent Research, Natural Sciences (DFF-4181-00196) and L.B.O. acknowledges support from the Danish National Research Foundation (DNRF116). M.R.A. and J.N. acknowledge support from the Novo Nordisk Foundation (NNF18OC0034936). A.D. acknowledges support from the Novo Nordisk Foundation (grant No. NNF18SA0035142), Villum Fonden (Grant no. 29476), Danish Council for Independent Research, Natural Sciences (DFF-117155-1001), and funding from the European Union's Horizon 2020 research and innovation program under the MarieSklodowska-Curie grant agreement No. 847523 (INTERACTIONS). J.T.E. acknowledges support from the Novo Nordisk Foundation (Hallas Møller Stipend). We thank Joshua Brickmann for providing us with embryonic stem cells, Anne Benedicte Mengel Pers for providing hepatocytes and Agnieszka Kawska for help with schematic illustrations.

## Author contributions

P.M.B. initiated and supervised the study. N.L. performed optical trapping and imaging experiments and performed all statistical analysis, Y.F.B. performed all tracking of filopodia, S.L.S., M.R.A. carried out silencing of cells, M.R.A., Y.F.B., N.L. conducted imaging of silenced cells and control cells, B.V. and N.L. performed tether pulling experiments, L.W. and N.L. carried out imaging of cells embedded within collagen, S.S. purified plasmids for imaging of actin, L.B.O. supervised optical trapping and provided optical tweezers setup, A.D. carried out active matter simulations, N.L., P.M.B, A.D. wrote the first version of the manuscript. Conceptualization of research: P.M.B, A.D., L.B.O., J.N., J.T.E., all authors read and approved the manuscript for publication.

## Competing interests

The authors declare no competing interests.
