## [Peer Review File · Nature Communications]

REVIEWER COMMENTS

Reviewer #1 (Remarks to the Author):

This manuscript by Leijnse et al. reports on dynamics of filopodia that are mechanically generated and modulated by optical tweezers. Various filopodia dynamics were observed in live-cell imaging, including helical rotation, tip movement and buckling. The authors attributed these observations to twisting motions of actin filaments inside the filopodia, and replicated these findings in both stem cells with pluripotency and terminally differentiated cells, claiming generality of their observation. These observations are interesting and likely of importance for understanding filopodia dynamics. There are however several questions, in particular, related to application of the optical tweezers, that need to be addressed before recommendation of the manuscript for publication in Nature Communications.

Major comments

1. In figure 2, after generating a protruded structure with pulling optical tweezers, the author observed helical rotatory motions for smaller beads attached to the filopodia membranes. Since these smaller beads (diameter $\sim 1 \mu\text{m}$) were coated with vitronectin that bind to the integrin receptors, the authors concluded that the observed helical motions were due to actins' twisting motion, which constitutes one of the major conclusions of the manuscript. These smaller beads, however, still have a micrometer-size diameter and should thus be prone to non-specific binding to the cell membranes (for example, as demonstrated in Cell (2016) 165, 1507). It will be interesting to see whether such helical motions disappear for other same-sized beads that bind to different membrane proteins present in the filopodia but not interacting with actins. If the helical motions persist for these beads, that would mean the entire filopodia membranes show collective helical movement (i.e., lipidic flow). In this case, orthogonal evidence would be desirable to support the authors' hypothesis that actin filaments drive the helical motions. One such possible experiment is to examine the helical motions of smaller beads upon addition of chemical reagents that intervene with actin filament dynamics.

2. In Figure 5, the authors determined the traction force of single filopodia using optical tweezers. Since the entire cells could be in motion during the optical tweezer measurement (stem cells have an inherent ability to migrate), the authors should clarify the effect of cell motility on their determination of the filopodia contraction force.

2-1. A related question is whether multiple tethers were involved in measuring the contraction force. Because of the micrometer size of the beads used for pulling, there could be formation of multiple tethers (i.e., one single bead was connected to multiple actin filaments within a filopodium), which make account for broad

distributions observed for the contraction force (Figure 5F).

3. The interesting observation that filopodia show buckling motions even under high pulling force suggests that the actins' twisting motions, rather than membrane elasticity, are responsible for the observed buckling. The authors further claim that the external resistance to rotation imposed by collagen matrixes leads to enhanced helical bucking of the filopodia (on page 9, the second paragraph) but without direct evidence present in the current manuscript. Do the authors indeed observe more frequent buckling as cells are embedded in more stiff collagen matrixes?

Typos

Figure1 legend, 12th line: vitronectin coated $d = 4.95 \mu\text{m}$ bead

Figure2 legend, 4th line: bead (grey, $d = 4.95 \mu\text{m}$).

Reviewer #2 (Remarks to the Author):

Filopodia rotate and coil by actively generating twist in their actin shaft by Leijnse et al.

In this work the authors explore dynamics of filopodia, focusing particularly on rotation and twisting dynamic properties. The authors find that filopodia explore their 3D extracellular space by combining growth and shrinking with axial twisting and buckling of their actin rich core. Interestingly, the authors show that the rotational dynamics of the filamentous actin inside filopodia is rather general phenomena, which can be observed for a range of highly distinct and cognate cell types spanning from earliest development to highly differentiated tissue cells. The authors use optical trapping to pull tethers (into which actin polymerize to form a filopodia), bind beads, and extract the forces generated by the growing/shrinking/rotating filopodia. These measurements show that filopodia exert traction forces and form helical buckles in a range of different cell types that can be ascribed to accumulation of sufficient twist.

In order to explain the possible origin of their findings, and notably, the emergence of filopodia rotation and twisting, the authors develop a non-equilibrium physical modeling of actin and myosin, demonstrating that twisting, and hence rotation, is an emergent phenomenon of active filaments confined in a narrow channel.

Based on their combined experimental/physical modeling results, the authors conclude that 'activity' induced twisting of the actin shaft is a general mechanism underlying fundamental functions of filopodia.

I find the experimental work beautiful and elegant. The physical model, in contrast, although nice by itself, I don't understand how it explains the experiments. The model is based on the classical hydrodynamic theories of active gels. The authors use this approach to show how confinement, activity, and elasticity control filopodia dynamics. In their model, the authors state that the active stress (force dipoles) accounts for the local stresses generated by active processes, including actomyosin polymerization and contractility.

- I don't see where there is myosin contractility along the actin shaft within the filopodia (except at the base which induce retraction of the all filopodia structure).
- Also, as far as I know, polymerization occurs essential at the tip and is not a local property. So, what are the origin of the force dipoles within the filopodia.
- The authors mention the presence of Myosin V and X within filopodia. Yet, in order to generate torque, the motors need to interact with multiple filaments at once (at least two). Myosin V and X do not (like myosin II) assemble to form aggregates, they essentially walk along the filament. Thus I don't see how force dipoles are generated by these motors.
- In fact, rotation and twisting of actin filaments is a property of formin mdia1 (and maybe others) and this alone could generate the twisting the authors measure (see papers by Kozlov and Bershadsky (2004,2005) and recent experiments by Yu et al. Nat. Comm 2017). The authors did not consider this possibility at all and in fact should check it.

In order to check their data and discriminate between possible mechanisms leading to rotation and twisting, the authors should perform the following control experiments.

1. Knockdown myosin X and V and check if rotation and twisting is affected).
2. Check the present of formins, notably mdia1, and see if knockdown of any of the various formins affects or eliminate rotation/twisting.

Reply to referees - Filopodia rotate and coil by actively generating twist in their actin shaft

Natascha Leijnse^{**},[†] Younes Farhangi Barooji^{**},[†] Mohammad Reza Arastoo,[†]
Stine Lauritzen Sønder,[‡] Bram Verhagen,[†] Lena Wullkopf,[¶] Janine Terra Eler,[¶]
Szabolcs Semsey,[†] Jesper Nylandsted,[§] Lene Broeng Oddershede,[†] Amin
Doostmohammadi,^{*,†} and Poul Martin Bendix^{*,†}

[†]*Niels Bohr Institute, University of Copenhagen, 2100 Copenhagen, Denmark*

[‡]*Membrane Integrity, Danish Cancer Society Research Center, Strandboulevarden 49, 2100
Copenhagen, Denmark*

[¶]*Biotech Research and Innovation Centre (BRIC), University of Copenhagen, Ole Maaløes
Vej 5, 2200 Copenhagen, Denmark*

[§]*Membrane Integrity, Danish Cancer Society Research Center, Strandboulevarden 49, 2100
Copenhagen, Denmark and Department of Cellular and Molecular Medicine, Faculty of
Health Sciences, University of Copenhagen, Blegdamsvej 3C, 2200 Copenhagen, Denmark*

E-mail: theory:doostmohammadi@nbi.ku.dk; experiments:bendix@nbi.ku.dk

Reviewer comments

Please find below our answers to the two reviewers. As a major change we have made new silencing studies to gather evidence for the activity induced mechanism underlying the rotations. The new results are plotted in the new Fig. 3j and the movement is plotted as a mean angular velocity. To keep consistency with Fig. 3i we have also quantified the existing data as angular velocities in the new Fig. 3i in the revised manuscript. The bead rotations in Fig. 2 and Supplementary Appendix are still presented as frequencies.

Reviewer 1 (Remarks to the Author)

This manuscript by Leijnse et al. reports on dynamics of filopodia that are mechanically generated and modulated by optical tweezers. Various filopodia dynamics were observed in live-cell imaging, including helical rotation, tip movement and buckling. The authors attributed these observations to twisting motions of actin filaments inside the filopodia, and replicated these findings in both stem cells with pluripotency and terminally differentiated cells, claiming generality of their observation. These observations are interesting and likely of importance for understanding filopodia dynamics. There are however several questions, in particular, related to application of the optical tweezers, that need to be addressed before recommendation of the manuscript for publication in Nature Communications.

Response We thank the reviewer for the careful reading and insightful comments and appreciate that the reviewer finds our work important for understanding filopodia dynamics. In the following we have elaborated on the experiments with optical tweezers and beads attached to filopodia. Changes in the manuscript are highlighted with a red font.

Major comments Reviewer 1

Comment 1 In Fig. 2, after generating a protruded structure with pulling optical tweezers, the author observed helical rotatory motions for smaller beads attached to the filopodia membranes. Since these smaller beads (diameter $\sim 1 \mu\text{m}$) were coated with vitronectin that bind to the integrin receptors, the authors concluded that the observed helical motions were due to actins' twisting motion, which constitutes one of the major conclusions of the manuscript. These smaller beads, however, still have a micrometer-size diameter and should thus be prone to non-specific binding to the cell membranes (for example, as demonstrated in Cell (2016) 165, 1507). It will be interesting to see whether such helical motions disappear for other same-sized beads that bind to different membrane proteins present in the filopodia but not interacting with actins. If the helical motions persist for these beads, that would mean the entire filopodia membranes show collective helical movement (i.e., lipidic flow). In this case, orthogonal evidence would be desirable to support the authors' hypothesis that actin filaments drive the helical motions. One such possible experiment is to examine the helical motions of smaller beads upon addition of chemical reagents that intervene with actin filament dynamics.

Reply 1 The beads we use are indeed prone to bind non-specifically to the plasma membrane as suggested by the reviewer and mentioned in the reference from the reviewer, Ref.¹ However, specific linking of the bead to other proteins would not necessarily prevent interactions between the bead and actin (e.g. through nonspecific interaction with integrins) and hence bead rotations originating from a spinning actin shaft would persist. Beads, which are pushed against a filopodium with an optical trap, are well known to bind non-specifically to filopodia membranes as shown in Ref.² (and also evident from our own experiments). The non-specifically bound beads do in fact connect to the actin since they travel along the filopodium with the actin retrograde flow, as shown for surface attached filopodia in Ref.² Instead of using an optical trap, another possibility would be to let beads bind spon-

taneously to target proteins on the filopodia. However, this strategy is problematic since only very few beads bind spontaneously to the small area on the filopodium and beads are quickly transported along the filopodium towards the cell body and are hence depleted from the filopodium. An optical trap is therefore needed to add a bead near the tip just prior to tracking its motion. This bead becomes bound specifically or non-specifically to the filopodium. This means that a bead will likely be attached to some actin interacting proteins and hence the bead will be linked to the actin structure. The reason for using vitronectin on our beads is merely to increase the likelihood that the bead is connected to integrin.

Regarding the helical flow of lipids mentioned by the reviewer, we are not familiar with such a mechanism and if such a flow should exist on a filopodium it is unlikely to occur without the participation of the actin. Interestingly, we can rarely observe beads moving randomly when not properly attached to the actin shaft, see Fig. 1 below. Such observations clearly show that binding to the membrane only (and not to the actin) leads to random Brownian diffusion and does not lead to directed rotations and retrograde movement of the bead.

Another evidence for the involvement of the actin is that we also observe rotations of our filopodia while they are kinked at the middle region, see e.g. Fig. 1B,D and Fig. 3A in the manuscript. Here we observe that the tip explores the 3D extracellular space in a rotary fashion. This motion must involve circular movement of the actin structure.

Interfering with the actin dynamics (like polymerization) is tricky since depolymerization of F-actin will make filopodia disappear. However, to gather some evidence for the mechanism behind the observed rotations, we have interfered indirectly with the actin dynamics by carrying out silencing experiments of myosin V and myosin X which resulted in reduced rotations when one of these motors is silenced (see answer to comment 4, Reviewer 2). This strongly indicates involvement of actin in the rotation of filopodia.

We have expanded the discussion regarding the binding of the beads to filopodia in the revised manuscript and added the two mentioned references.

New text (red font): We explicitly performed 3D tracking of the whole filopodium and its tip, see Fig. 3 and 4 in the manuscript, and Supplementary Figs. 1,7,8,16 and made complimentary measurements of the spinning of the actin shaft within the filopodium, see Fig. 2 in manuscript and Supplementary Figs. 4-6. Beads were added to the side of the filopodium by an optical trap and were subsequently tracked in 3D. Vitronectin on the beads allows for binding directly to integrin which can couple the bead to the internal actin structure. We note that beads can also bind non-specifically to plasma membranes¹ or to filopodia and still be coupled to the actin structure, as shown in Ref.² A clear signature that the bead is connected to the actin is translocation of the bead towards the cell body which is caused by retrograde flow within the filopodium (Fig. 2a-c and Supplementary Fig. 4). The results from these experiments showed that the actin shaft has the ability to spin with a similar angular velocity as the circular movement of the whole filopodium. These results strongly suggest that the spinning of the actin together with filopodia bending and growth, is responsible for the 3D motion and buckling of filopodia.

Comment 2 In Fig. 5, the authors determined the traction force of single filopodia using optical tweezers. Since the entire cells could be in motion during the optical tweezer measurement (stem cells have an inherent ability to migrate), the authors should clarify the effect of cell motility on their determination of the filopodia contraction force.

Reply 2 This is a really good point, which is addressed as follows: The cells we are measuring on are indeed motile as the reviewer suggests. However, the cell movement is quite slow compared to the force kinetics we measure. From confocal imaging we have tracked two random cells while also measuring the forces exerted by the filopodium tip, see Fig. 2, below. The position of the cell was determined by tracking the junction between the cell and the filopodium/cell-tether while the force was detected by using the photo diode. As seen

in Fig. 2 below, there is no clear correlation between the slow movement of the cell and the fast kinetics seen in the force curve.

As seen in Fig. 2 below, the tracked cells move less than $1\mu\text{m}$ during 200 s. It is known that actin polymerization elongates the filopodium at a speed of $\sim 100\text{-}200\text{nm/s}$ (see e.g. Fig. 3C in Supplementary Appendix in Ref.³). Therefore, we expect actin dynamics within the filopodium to be responsible for the observed changes in the force.

We have added Fig. 2 below to the Supplementary Information to show that the filopodia-force and cell-movement are decoupled.

New text in subsection "Filopodia buckle and pull at the same time": To exclude a contribution from the cell motility present in most cell types, we performed parallel tracking of cells and force measurements. As seen in Supplementary Fig. 14 the cell movement is slower than the rapid changes in the force and are therefore uncoupled. The measured force can therefore be ascribed to pulling by the filopodium.

Comment 3 A related question is whether multiple tethers were involved in measuring the contraction force. Because of the micrometer size of the beads used for pulling, there could be formation of multiple tethers (i.e., one single bead was connected to multiple actin filaments within a filopodium), which make account for broad distributions observed for the contraction force (Fig. 5F).

Reply 3 It can indeed occur that multiple tethers are extracted. However, we are able to detect multiple tethers during confocal imaging. Fig. 3 below shows an example of multiple tethers extracted from a HEK293 cell (transfected with GFP-UtrCH which labels F-actin) using an optically trapped $d=4.95\ \mu\text{m}$ bead and the corresponding force curve. The bead was pushed onto the cell membrane to create binding and then translated away with a constant velocity of $0.005\ \mu\text{m/s}$. Numbers 1-4 in the force curve correspond to the confocal images

on the right, where multiple tethers are visible (red arrows) and are breaking off during the course of the experiment. Those multiple tethers were visible both, in the confocal images as well as in the force curve as shown in Fig. 3 below.

Comment 4 The interesting observation that filopodia show buckling motions even under high pulling force suggests that the actins' twisting motions, rather than membrane elasticity, are responsible for the observed buckling. The authors further claim that the external resistance to rotation imposed by collagen matrixes leads to enhanced helical buckling of the filopodia (on page 9, the second paragraph) but without direct evidence present in the current manuscript. Do the authors indeed observe more frequent buckling as cells are embedded in more stiff collagen matrixes?

Reply 4 We thank the reviewer for pointing this out- we do indeed indicate that the extracellular matrix leads to more buckling which is a quite intuitive scenario, although not supported by our data. However, there is no reason to doubt the existence of friction between filopodia and the external matrix which constitutes a target for filopodia during e.g. migration and invasion.⁴ The purpose was to argue that an external resistance, which could naturally originate from a collagen matrix (both low and high stiffness), could contribute to build-up of twist in the rotating actin shaft. This would require interaction between the filopodia and the external environment which is quite frequent for living cells. The expected increase in buckling behavior, in presence of an extracellular matrix, is simply a logical argument since any resistance (e.g. friction) to rotations will evidently enhance build-up of twist in a rotating structure and hence further amplify the buckling behavior.

We have reformulated this to avoid any misunderstanding:

New text (red font): The presence of an extracellular matrix could contribute to build-up

of twist in the rotating filopodia. Specific and unspecific interactions between filopodia and the surrounding matrix will lead to twist in the rotating structure and further induce buckling in the case that sufficient twist is accumulated. Filopodia in cells, migrating in a 3D collagen I network, are expected to experience external friction at the contact points between filopodia and the fibers. Additionally, friction is likely to exist between the plasma membrane and the actin shaft mediated by various proteins linking the membrane with the actin. The helical buckles and coils observed in manuscript Figures 4B, C and D, Supplementary Figs. 3, 16 and movie 4b can hence be a signature of over-twisting of the actin shaft which occurs naturally when rotating filopodia interact with a collagen I network or the membrane.

Comment 5 - Typos Fig. 1 legend, 12th line: vitronectin coated $d = 4.95 \mu\text{m}$ bead
Fig. 2 legend, 4th line: bead (grey, $d = 4.95 \mu\text{m}$ bead).

Reply 5 These typos have been corrected and we thank the reviewer for careful reading of our manuscript.

Reviewer 2 (Remarks to the Author)

Filopodia rotate and coil by actively generating twist in their actin shaft by Leijnse et al. In this work the authors explore dynamics of filopodia, focusing particularly on rotation and twisting dynamic properties. The authors find that filopodia explore their 3D extracellular space by combining growth and shrinking with axial twisting and buckling of their actin rich core. Interestingly, the authors show that the rotational dynamics of the filamentous actin inside filopodia is rather general phenomena, which can be observed for a range of highly distinct and cognate cell types spanning from earliest development to highly differentiated tissue cells. The authors use optical trapping to pull tethers (into which actin polymerize to form a filopodia), bind beads, and extract the forces generated by the growing/shrinking/rotating filopodia. These measurements show that filopodia exert traction forces and form helical

buckles in a range of different cell types that can be ascribed to accumulation of sufficient twist.

In order to explain the possible origin of their findings, and notably, the emergence of filopodia rotation and twisting, the authors develop a non-equilibrium physical modeling of actin and myosin, demonstrating that twisting, and hence rotation, is an emergent phenomenon of active filaments confined in a narrow channel.

Based on their combined experimental/physical modeling results, the authors conclude that ‘activity’ induced twisting of the actin shaft is a general mechanism underlying fundamental functions of filopodia.

I find the experimental work beautiful and elegant. The physical model, in contrast, although nice by itself, I don’t understand how it explains the experiments. The model is based on the classical hydrodynamic theories of active gels. The authors use this approach to show how confinement, activity, and elasticity control filopodia dynamics. In their model, the authors state that the active stress (force dipoles) accounts for the local stresses generated by active processes, including actomyosin polymerization and contractility.

Response We appreciate that the reviewer finds our results interesting and the experiments to be both elegant and beautiful. With respect to the proposed force dipoles in the model we have elaborated on our assumptions for the model below and in the revised manuscript. Changes to the manuscript have been highlighted with a red font.

Comment 1 I don’t see where there is myosin contractility along the actin shaft within the filopodia (except at the base which induce retraction of the all filopodia structure) Also, as far as I know, polymerization occurs essential at the tip and is not a local property. So, what are the origin of the force dipoles within the filopodia.

Reply 1 We thank the referee for this comment, which made us realise that we have not

been sufficiently clear in explaining the force and torque generation. A classical view of actin organization in filopodia suggested that actin filaments form stable bundles along the filopodia shaft and polymerization and contraction occur at the tip and base of filopodia, respectively. More recent analyses of actin filaments microstructure within the filopodia has, however, revealed that some additional actin reorganization through various capping proteins occurs all along the actin shaft.⁵⁻⁷ Moreover, the role of myosin X and myosin V motor proteins in filopodia dynamics is well-established and these motors have been found in neuronal, cancer and other cell types.⁸⁻¹² As shown below we also confirmed that silencing of myosin Va,b and myosin X has an effect on rotational behavior of filopodia in MCF7 cells. We indeed argue that the mere walking motion of these motor proteins along actin filaments with helical structures can produce force and torque dipoles as shown schematically by the new Fig. 7a (Fig. 4 below). Both of these motors experience a drag while walking along the actin bundle- myosin V from dragging vesicle encapsulated cargo while myosin X is exposed to drag while transporting membrane embedded proteins along the viscous plasma membrane as it walks towards the tip of the filopodium. It is worth noting that since we use a continuum model to represent the dynamics of actin filaments within the filopodia, we are accounting for a coarse-grained impact of actin reorganisation rather than the details of the force generation mechanism. Within this framework, it is well-established that to the lowest order the impact of any non-equilibrium effect can be accounted to by active stresses that represent force and torque dipoles.¹³ As such, our model provides a generic prediction on the role of lowest order non-equilibrium forces within the filopodia in inducing helical motion of the filopodia.

We have elaborated on the mechanism behind activity within filopodia in the revised manuscript and also added Fig. 7a as a new Fig. in the manuscript.

In section: **Twist deformations are a generic non-equilibrium feature of confined**

actomyosin complexes

New text: The existence of molecular activity in filopodia is well established through the presence of actin reorganization proteins^{5,7,14} and molecular motors such as myosin V and myosin X.⁸⁻¹²

In section: **Discussion**

New text (red font): Our model assumes active force generation along the actin shaft which could arise from walking motion of myosin motor proteins along actin filaments with helical structures. The presence of myosin V and myosin X in filopodia of cancer cells and other cell types is well established⁸⁻¹² and these motors can produce force and torque dipoles as shown schematically in Fig. 7a. Both of these motors experience a drag while walking along the actin bundle: myosin V from dragging vesicle encapsulated cargo while myosin X causes drag from transporting membrane embedded proteins along the viscous plasma membrane as it walks towards the tip of the filopodium. Indeed we measured reduction in filopodia rotations upon silencing myosin V or myosin X which supports the force generating ability of these motors on the actin structure. The induced chirality of the rotations is random, but can be biased by active torque dipoles in the system. Such dipoles indeed exist in natural filopodia in the form of myosins exhibiting chiral motion and inter-filament stepping along actin bundles as demonstrated for the filopodia associated motors myosin V¹⁵ and myosin X.¹⁶ Both of these motors walk towards the tip in a filopodium and spiral around the actin in a clockwise orientation as seen from the tip thus inducing counterclockwise rotations, as seen from the perspective of the tip. Our experimental data indeed show a preference for counterclockwise orientation of the twist whereas mRNA silencing of myosin V and myosin X resulted in increased randomness for the orientation (Supplementary Table 3) thus supporting the idea that torque-generation by these motors induces a bias

towards a specific chirality of the rotations. The presence of both twist-orientations supports the theoretical predictions of a hydrodynamic instability being the origin of the twist.

Comment 2 The authors mention the presence of Myosin V and X within filopodia. Yet, in order to generate torque, the motors need to interact with multiple filaments at once (at least two). Myosin V and X do not (like myosin II) assemble to form aggregates, they essentially walk along the filament. Thus I don't see how force dipoles are generated by these motors.

Reply 2 This is a good point raised by the reviewer - how do these motors lead to a torque in the actin bundle? Myosin X motors interact with the plasma membrane whereas myosin V transports cargo vesicles as shown by the new schematic in new Fig. 7A in the revised manuscript. Friction between the plasma membrane, or cytosol, and these motors therefore allows these motors to exert a force on the actin as argued in the response to comment 1. Additionally, these motors have been shown to walk in a spiral path along actin bundles by performing interfilament stepping. This was reported in Ref.^{15,16} where single myosin motors (tracked by QDs linked to their tail) were shown to undergo spiral motion around bundles of actin. The step size for myosin V and X on these bundles differed from the step size on single filaments which supports interfilament stepping by these motors. The local force and spiral path followed by these myosins would then generate torque on the actin structure which could bias the preferred orientation of the helical twist. Indeed our silencing experiments, discussed in the response to comment 4 below, show that in addition to a lower mean angular velocity the silencing also leads to a more random twist orientation (Supplementary Table 3) which is supported by our model.

In the revised manuscript we have mentioned the reduction in angular velocity and increased randomness of the rotations following silencing of myosin V and myosin X and discussed the torque generation, see answers to comment 1 and 4 for reviewer 2.

Comment 3 In fact, rotation and twisting of actin filaments is a property of formin mdia1 (and maybe others) and this alone could generate the twisting the authors measure (see papers by Kozlov and Bershadsky (2004,2005) and recent experiments by Yu et al. Nat. Comm 2017). The authors did not consider this possibility at all and in fact should check it.

Reply 3 mDia1-3 formins indeed have crucial roles in filopodia function in many cells, but also during formation of filopodia.¹⁷⁻¹⁹ Due to the role of mDia2 in filopodia formation it may be problematic to silence this protein for studying filopodia due to the absence of filopodia after efficient silencing. In this context it has been reported that silencing mDia2 lead to cells with few or no filopodia.¹⁸ However, we have made new controls and expressed fluorescent mDia1-mEmerald (another formin) together with Lifeact-mCherry to detect the possible presence of mDia1 in MCF7 cells. We used co-transfection of fluorescent actin to detect filopodia and they were still observed to be present after silencing of mDia1. Although some mDia1 was visible in filopodia, this protein was predominantly detected in the cytoplasm as shown in Fig. 5 below, and in most cells the signal in the filopodium was extremely weak.

Despite the low levels of mDia1 in filopodia from MCF7 cells we tested the effect of silencing mDia1 (Fig. 6 below) on rotation and buckling as requested by the reviewer. With a silencing efficiency of 95-99% using short interfering RNA (siRNA) we still detected similar mean angular velocities as in cells transfected with control siRNA, see Fig. 7 below.

However, as mentioned by the reviewer, formins have been shown to sense force and torque in actin and to regulate polymerization.²⁰⁻²² Therefore, we have briefly discussed the role of mDia in the revised manuscript and added Figs. 5 and 7 below as Supplementary Figures and added the two mentioned references by Kozlov et al. 2004 and Yu et al. 2017 to the revised manuscript.

New text in subsection "Myosin activity affects filopodia rotation":

Another molecular component important for filopodia function is the formin mDia1. mDia1 has been reported to sense twist in actin filaments²⁰⁻²² and hence could be involved in twisting the actin shaft of filopodia. Fluorescently tagged mDia1 was found to be present in MCF7 cells and a slight signal was also detected within the filopodia of MCF7 cells (see Supplementary Fig. 10). We therefore performed a silencing study on mDia1 to investigate any possible effect on filopodia rotation. As shown in Fig. 12 there was not a significant change in the mean angular velocity following silencing of mDia1.

Overall, these results strongly suggest that molecular activity from myosin V and myosin X are somehow involved in the rotation of filopodia whereas the actin binding protein mDia1 does not play a role in the observed rotations.

Comment 4.1 In order to check their data and discriminate between possible mechanisms leading to rotation and twisting, the authors should perform the following control experiments.

1. Knockdown myosin X and V and check if rotation and twisting is affected.

Reply 4.1

We have carried out silencing of both myosin V and X (Fig. 8 below), as requested by the reviewer, and analyzed the resulting mean angular velocities. As shown in Fig. 9 below and new Fig. 3J in the revised manuscript, we measure a significant decrease in angular velocity after silencing myosin Va and Vb and a slightly less, but still significant effect of silencing myosin X.

Interestingly, we still observe rotation and helical buckles in filopodia after silencing these myosins. These results clearly indicate that rotation and helical twisting are not solely caused by the activity of one of these motors alone, but could be due to the combined activity from several motors and also polymerization along the shaft, as suggested by our model. The orientational bias observed in the rotation was also reduced after silencing myosin, see Sup-

plementary Table 3, which is consistent with the fact that myosin Va walks counterclockwise around actin bundles (away from the cell) and hence generates a torque in the opposite direction. The combined reduction in mean angular velocity and increased randomness of the orientation strongly indicates that the activity of these motors plays a role in the observed filopodia rotations.

We have added the following new subsection in the revised manuscript to discuss the effects of silencing myosin V and X:

Myosin activity affects filopodia rotation

To shed light on the active mechanism leading to filopodia rotations we next performed silencing of genes encoding for molecular activity within the filopodia. In particular myosin V and myosin X motors have been reported to be associated with filopodia formation and activity and have been shown to transport membrane proteins and vesicular content along the filopodium^{8-12,23}.

We performed mRNA silencing of myosin Va,b and myosin X which led to significant reduction of the expression of these motors in MCF7 cells, see new Supplementary Fig. 9 in the revised Supplementary Information. By expressing EGFP Lifeact in MCF7 enabled us to track the filopodia in cells silenced for myosins and extract the mean angular velocity. The results showed a significant reduction in the mean angular velocity of filopodia when myosin Va silencing is compared to cells which were transfected with control siRNA (CSI), see Fig. 3j. Furthermore, we also measured a reduction in the mean angular velocity when silencing myosin Vb or myosin X when compared to the control experiments (CSI), see Fig. 3j in the revised manuscript. Also, the chirality of the rotations was affected by the silencing of myosin activity (see Supplementary Table 3) leading to larger degree of randomness after

silencing.

Another molecular component important for filopodia function is the formin mDia1. mDia1 has been reported to respond to twist in actin filaments²⁰⁻²² and hence could be involved in twisting the actin shaft in filopodia. Fluorescently tagged mDia1 was found to be present in MCF7 cells and a slight signal could also be detected from the filopodia of MCF7 cells (see Supplementary Fig. 10). We therefore performed a silencing study on mDia1 (Supplementary Fig. 11) to investigate any possible effect on filopodia rotation. As shown in Supplementary Fig. 12 there was not a significant change in the mean angular velocity following silencing of mDia1.

And in **Conclusion** we added the following text: Filopodia rotations were reduced by silencing either myosin V or myosin X and the chirality of the rotations became less biased. The activity from these motors, which walk in a helical path around actin bundles, is therefore responsible for spinning or twisting the actin core of filopodia.

2 Check the present of formins, notably mdia1, and see if knockdown of any of the various formins affects or eliminate rotation/twisting.

Reply 4.2

This issue was addressed under **Comment 3** above.

The following Materials and Methods have been added as a result of the revision.

Control experiments

siRNAs and transfection

Myosin Va, Vb, X, mDia1 and control (CSI) siRNAs were purchased from Addgene and efficiency was tested with western blots, see Fig. 6 and Fig. 8.

Reverse siRNA transfections of MCF7 cells were performed using transfection reagent

(Invitrogen) with 25 nM siRNA (Sigma-Aldrich) according to manufacturer's protocol. The siRNA containing medium was replaced after 24 h and the analysis performed after 72 h.

SiRNAs were purchased from Sigma Aldrich unless otherwise stated. SiRNA sequences:

Control siRNA (AllStar Negative Control siRNA, 1027281 Qiagen)

MYO5A#1 siRNA (5'-CUGACUACCUGAAUGAUGA-3'),

MYO5A#2 siRNA (5'-CGAAACAACUGGAACUCGA-3'),

MYO5B#1 siRNA (5'-GACAUAGAUUUGGACCCGA-3'),

MYO5B#2 siRNA (5'-GAGAUCAUCCUGCAGGUAU -3'),

MYO10#1 siRNA (5'-CUUACGAAUCUCUUAAGAA-3'),

MYO10#2 siRNA (5'-GAAUCAGUCUGGAUGUGUA-3'),

mDia1#1 siRNA (5'-CAUGUGAGGAGUUACGUAA-3'),

mDia1#2 siRNA (5'-GACAGAAGAAGGAAUCCUA-3')

For live cell control experiments, siRNA transfections were performed using Oligofectamine transfection reagent and cells were kept in a humidified incubator at 37 °C and 5 % CO₂. The culture medium was renewed after 24 h.

Plasmid transfection

48 h post RNAi, the cells were transiently transfected with the plasmid(s) of interest (Lifeact GFP plasmid, Lifeact-mCherry plasmid, a kind gift from Roland Wedlich Söldner; mEmerald-mDia1-C-14, a kind gift from Michael Davidson, Addgene plasmid # 54156) using Lipofectamine LTX according to the manufacturer's protocol. Briefly, 1.25 µg plasmid was diluted in 500 µl OptiMem and 5 µl Lipofectamine LTX was added and incubated at RT for 25 min. After washing the cells with PBS the transfection mixture was added. The volume was adjusted with OptiMem and cells were kept in the humidified incubator at 37 °C and 5 % CO₂. After 2 h 45 min, the medium was replaced with DMEM supplemented with 10 % FBS and 1 % PenStrep.

Immunoblotting

Cells were lysed in Laemmli sample buffer (125 mM Tris, pH 6.7, 140 mM SDS, 20 % glycerol, 0.3 μ M bromophenol blue) supplemented with protease inhibitor cocktail (Roche 4693124001), phosphatase inhibitor (Roche 4906837001) and 0.1 M dithiothreitol (DTT). Cell lysates were boiled for 5 min and separated by SDS-PAGE using precast 4-15 % gradient gels (BioRad) followed by transfer to nitrocellulose membranes (BioRad) using Trans-Blot Turbo™ transfer system and blocked in PBS, containing 0.1 % Tween-20 (PBST) and 5 % BSA. The molecular weights of proteins from the gels were evaluated using Novex™ Sharp Protein Standard (Invitrogen). Membranes were incubated with primary antibodies in PBST/5 %BSA at 4°C overnight (mDia1 1:500 dilution, BD Transduction Laboratories 610848; Myosin 5a, 1:1000 dilution, Cell Signaling Technology 3402; Myosin 5b, 1:500 dilution, Novus Biologicals NBP1-87746; Myosin 10, 1 μ g/mL, Sigma Aldrich HPA024223; Hsp90, BD Transduction Laboratories 610418, 1:4000, GAPDH, 1:7500 dilution, Abcam ab189095). Membrane was washed followed by incubation with appropriate peroxidase-conjugated secondary antibodies (DAKO) for 0.5 h at RT. Chemiluminescent signals (Clarity Western ECL substrate, BioRad) were detected with Luminescent Image Reader (LAS-1000Plus, Fujifilm).

Sample preparation for imaging

On the day of imaging, the transfected cells were detached from the 6-well plate by pipetting and part of the cell suspension transferred to a Mattek glass bottom dish were incubated for 5 h to have cells attached to the glass surface of the dish. Prior to imaging, the culture medium was replaced with FluoroBrite DMEM.

Control Figures and Tables

Table 1: Orientation of filopodia rotations before and after silencing of myosin Va, Vb, X, and mDia1 (CW (clockwise), CCW (counterclockwise)). Both experiments are compared with samples using control siRNA. MCF7 cells were transfected with Lifeact GFP to visualize the filopodia.

Experiment 1	N_{Filo}	CW (%)	CCW (%)	Random (%)	(mean ang. vel. \pm std) (deg/s)
Control siRNA	31	23	45	32	0.30 ± 0.23
mDia1 siRNA	35	20	54	26	0.21 ± 0.12
MyoVa siRNA	23	22	26	52	0.11 ± 0.07
MyoVb siRNA	28	4	36	61	0.13 ± 0.15
MyoX siRNA	18	6	50	44	0.16 ± 0.14
Experiment 2					
Control siRNA	61	11	51	38	0.28 ± 0.22
MyoVa siRNA	65	14	18	68	0.14 ± 0.17

Figure 1: Example of random motion of a bead which is attached to the filopodia membrane, but not linked to the actin shaft. (a) Color coded (blue to white) overlay of 293 consecutive confocal images of a VN coated $0.99 \mu\text{m}$ bead attached to the membrane of a filopodium (cyan, EGFP Lifeact-7) extracted from a MCF7 cell. The bead is attached to the filopodium but not linked to the actin at all times. Frame interval was 0.39 s and total time 114 s . Scale bar is $5 \mu\text{m}$. (b) 2D track of the bead movement from (a) where X denotes the direction along, and Y the direction perpendicular, to the filopodium. (c) X movement of the bead as a function of time illustrating its change in direction along the filopodium. The bead trajectory in (b) and (c) was obtained by manual blurring and thresholding of the image sequence followed by using the Fiji plugin TrackMate.

Figure 2: Tether force and cell movement. (a) Confocal image of a trapped bead (gray, reflection) holding a tether extracted from a cell (cyan, F-actin). Scale bar is 5 μm . The yellow rectangle marks the region for cell movement tracking described in (c). (b) Total holding force required to hold the bead attached to the tether shown in (a). Time in (b) and (c) denotes the time after onset of the experiment where the tether was extracted (extraction data not shown). (c) Quantification of the cell body movement (tracked inside the yellow rectangle in a) during the experiment in x and y direction.

Figure 3: Multiple tethers can be detected during confocal imaging. Force curve (left) and confocal images (right, cyan (F-actin via GFP-UtrCH) and gray (reflection) overlay, scale bar is 5 μm) corresponding to a tether pulling experiment from a HEK293 cell. Red arrows indicate tethers. After pushing the bead ($d=4.95 \mu\text{m}$) against the cell membrane, it was translated away with a constant velocity of 0.05 $\mu\text{m/s}$. Numbers 1-4 in force curve correspond to the confocal images on the right.

Figure 4: Schematic of how walking of myosin motor on actin bundles with inherent helical structures results in generation of force and torque dipoles. Both myosin X and V walk in a helical motion around actin bundles and have the ability to make interfilament steps.^{15,16} The drag experienced from the viscous membrane (myosin X) and also from the attached cargo (myosin V) results in local forces applied to the actin structure. Molecular force dipoles originating from other myosins present in the cell cortex have been described previously.²⁴

Figure 5: Confocal image of a MCF7 cell expressing (a) mEmerald-mDia1 (green) and (b) Lifeact-mCherry (red). (c,d) Enlarged view of the boxed region shown in (b).

Figure 6: (a) Immunoblot of lysates from MCF7 cells 72 h after siRNA transfection with indicated siRNAs (mDia1: 170 kDa, Hsp90: loading control). (b) Extended blot.

Figure 7: Results from silencing mDia1 in MCF7 cells on the angular velocities of individual filopodia compared with MCF7 cells transfected with control siRNA. There is no significant difference between the two populations (p -value = 0.18, two tailed Mann-Whitney test). Scatter plot shows the median and the whiskers extend from minimum to the maximum values. N (filopodia) = 92 (siRNA), 35 (mDia1).

Figure 8: (a) Immunoblot of lysates from MCF7 cells 72 h after siRNA transfection with indicated siRNAs (MYO5A: 207 kDa, MYO5B 214 kDa, MYO10 265 kDa, Hsp90: loading control). (b) Extended blots.

Figure 9: Mean angular rotation velocities of filopodia after silencing myosins Va, b and myosin X, respectively, were compared to cells exposed to control siRNA. Kruskal-Wallis test with significance set at $p < 0.05$. p -values are $p < 0.0001$, $p = 0.0003$, $p = 0.0171$, for control siRNA vs MyoVa, Vb, and X, respectively. Scatter plot shows the median and the whiskers extend from minimum to the maximum values and symbols describing p -values are *: $p < 0.05$; **: $p < 0.01$; ***: $p < 0.001$, ****: $p < 0.0001$. N (filopodia) = 92 (control siRNA), 88 (MyoVa), 28 (MyoVb), 18 (MyoX).

References

- (1) Seo, D.; Southard, K.; Kim, J.; Lee, H.; Farlow, J.; Lee, J.; Litt, D.; Haas, T.; Alivisatos, A.; Cheon, J. et al. A Mechanogenetic Toolkit for Interrogating Cell Signaling in Space and Time. Cell **2016**, 165, 1507–1518.
- (2) Kohler, F.; Rohrbach, A. Surfing along Filopodia: A Particle Transport Revealed by Molecular-Scale Fluctuation Analyses. Biophys. J. **2015**, 108, 2114–2125.
- (3) Leijnse, N.; Oddershede, L. B.; Bendix, P. M. Helical buckling of actin inside filopodia generates traction. Proc Natl Acad Sci U S A **2015**, 112, 136–41.
- (4) Jacquemet, G.; Hamidi, H.; Ivaska, J. Filopodia in cell adhesion, 3D migration and cancer cell invasion. Curr. Opin. Cell Biol. **2015**, 36, 23–31.
- (5) Medalia, O.; Beck, M.; Ecke, M.; Weber, I.; Neujahr, R.; Baumeister, W.; Gerisch, G. Organization of actin networks in intact filopodia. Current biology **2007**, 17, 79–84.
- (6) Breitsprecher, D.; Koestler, S. A.; Chizhov, I.; Nemethova, M.; Mueller, J.; Goode, B. L.; Small, J. V.; Rottner, K.; Faix, J. Cofilin cooperates with fascin to disassemble filopodial actin filaments. Journal of cell science **2011**, 124, 3305–3318.
- (7) Shekhar, S.; Kerleau, M.; Kühn, S.; Pernier, J.; Romet-Lemonne, G.; Jégou, A.; Carlier, M.-F. Formin and capping protein together embrace the actin filament in a ménage à trois. Nature communications **2015**, 6, 1–12.
- (8) Tamada, A.; Kawase, S.; Murakami, F.; Kamiguchi, H. Autonomous right-screw rotation of growth cone filopodia drives neurite turning. J Cell Biol **2010**, 188, 429–41.
- (9) Brawley, C.; Rock, R. Unconventional myosin traffic in cells reveals a selective actin cytoskeleton. PNAS **2009**, 106, 9685–9690.
- (10) Houdusse, A.; Titus, M. The many roles of myosins in filopodia, microvilli and stereocilia. Curr. Biol. **2021**, 31, R586–R602.

- (11) Lan, L.; Han, H.; Zuo, H.; Chen, Z.; Du, Y.; Zhao, W.; Gu, J.; Zhang, Z. Upregulation of myosin Va by Snail is involved in cancer cell migration and metastasis. Int. J. Cancer **2009**, 126, 53–64.
- (12) Li, Y.; Zhong, A.; Dong, H.; Ni, L.; Tan, F.; Yang, W. Myosin Va plays essential roles in maintaining normal mitosis, enhancing tumor cell motility and viability. Oncotarget **2017**, 8, 54654–54671.
- (13) Prost, J.; Jülicher, F.; Joanny, J. Active gel physics. Nature Physics **2015**, 11, 111 – 117.
- (14) Breitsprecher, D.; Koestler, S. A.; Chizhov, I.; Nemethova, M.; Mueller, J.; Goode, B. L.; Small, J. V.; Rottner, K.; Faix, J. Cofilin cooperates with fascin to disassemble filopodial actin filaments. J Cell Sci **2011**, 124, 3305–18.
- (15) Ali, M. Y.; Uemura, S.; Adachi, K.; Itoh, H.; Kinoshita, J., K.; Ishiwata, S. Myosin V is a left-handed spiral motor on the right-handed actin helix. Nat Struct Biol **2002**, 9, 464–7.
- (16) Sun, Y.; Sato, O.; Ruhnaw, F.; Arsenault, M. E.; Ikebe, M.; Goldman, Y. E. Single-molecule stepping and structural dynamics of myosin X. Nat Struct Mol Biol **2010**, 17, 485–91.
- (17) Ridley, A. Life at the leading edge. Cell **2011**, 145, 1012–1022.
- (18) Yang, C.; Czech, L.; Gerboth, S.; Kojima, S.; Scita, G.; Svitkina, T. Novel Roles of Formin mDia2 in Lamellipodia and Filopodia Formation in Motile Cells. PLoS Biol. **2007**, 11, 2624–2645.
- (19) Goh, W.; Lim, K.; Sudhaharan, T.; Sem, K.; Bu, W.; Chou, A.; Ahmed, S. mDia1 and WAVE2 Proteins Interact Directly with IRSp53 in Filopodia and Are Involved in Filopodium Formation. J. Biol. Chem. **2012**, 287, 4702–4714.

- (20) Kozlov, M.; Bershadsky, A. Processive capping by formin suggests a force driven mechanism of actin polymerization. J. Cell Biol. **2004**, 167, 1011–1017.
- (21) Yu, M.; Yuan, X.; Lu, C.; Shimin Le, S.; Kawamura, R.; Efremov, A.; Zhao, Z.; Kozlov, M.; Sheetz, M.; Bershadsky, A. et al. mDia1 senses both force and torque during F-actin filament polymerization. Nat. Commun. **2017**, 18.
- (22) Mizuno, H.; Tanaka, K.; Yamashiro, S.; Narita, A.; Watanabe, N. Helical rotation of the diaphanous-related formin mDia1 generates actin filaments resistant to cofilin. PNAS **2018**, 115, E5000–E5007.
- (23) Jacquemet, G.; Stubb, A.; Saup, R.; Miihkinen, M.; Kremneva, E.; Hamidi, H.; J., I. Filopodome Mapping Identifies p130Cas as a Mechanosensitive Regulator of Filopodia Stability. Curr. Biol. **2019**, 29, 202–216.
- (24) Naganathan, S. R.; Fürthauer, S.; Nishikawa, M.; Jülicher, F.; Grill, S. W. Active torque generation by the actomyosin cell cortex drives left–right symmetry breaking. Elife **2014**, 3, e04165.

REVIEWERS' COMMENTS

Reviewer #1 (Remarks to the Author):

The authors addressed most of this reviewer's concerns with additional experimental data and analyses. Thus, this reviewer recommends publication of the revised manuscript in Nature Communications.

Reviewer #2 (Remarks to the Author):

The authors have addressed my multiple questions/comments in their revised ms.